# Using GRACE in a streamflow recession to determine drainable water storage in the Mississippi River Basin

Heloisa Ehalt Macedo[1,2], Ralph Edward Beighley[2], Cédric H. David[3], John T. Reager[3]

[1]Geography Department, McGill University, Montreal, Quebec, Canada
[2]Civil and Environmental Engineering Department, Northeastern University, Boston, Massachusetts, USA
[3]NASA Jet Propulsion Laboratory, California Institute of Technology, Pasadena, California, USA

*Correspondence to*: Heloisa Ehalt Macedo (heloisa.ehaltmacedo@mail.mcgill.ca)

**Abstract.** The study of the relationship between water storage and runoff generation has long been a focus of the hydrological sciences. NASA's Gravity Recovery and Climate Experiment (GRACE) mission provides monthly depth-integrated information on terrestrial water storage anomalies derived from time-variable gravity observations. As the first basin-scale storage measurement technique, these data offer potentially novel insight into the storage-discharge relationship. Here, we apply GRACE data in a streamflow recession analysis with river discharge measurements across several subdomains of the Mississippi River Basin. Non-linear regression analysis was used for 12 watersheds to determine that the fraction of baseflow in streams during non-winter months varies from 52 to 75% regionally. Additionally, the first quantitative estimate of absolute drainable water storage was estimated. For the 2002-2014 period, the drainable storage in the Mississippi River Basin ranged from $2,900 \pm 400$ km$^3$ to $3,600 \pm 400$ km$^3$.

## 1 Introduction

The amount of water that a watershed stores is a key descriptor of the functionality of that watershed and its role in the Earth system (Wagener et al., 2007;Sayama et al., 2011;Black, 1997). As water can reside for periods ranging from months to thousands of years in subsurface soils, storage is often a critical yet under-observed variable in hydrology and rainfall-runoff models. Water storage helps to define the amount of water available for water resources applications, as well as the resilience of a watershed to changes in climate (eg., Brutsaert, 2005;Kirchner, 2009) with implications for society and the environment. Despite the importance of characterizing watershed storage, relatively little work has been done to understand the relationship between storage and discharge. Most of the existing work is based on remotely-sensed observations of storage (eg., Riegger and Tourian, 2014;Reager et al., 2014;Sproles et al., 2015;Tourian et al., 2018;Riegger, 2018). Across scales, subsurface heterogeneity in soils and geology can make the storage-discharge relationship complex and challenging to observe and model (Beven, 2006). Additionally, observations of storage over large domains such as an entire river basin are challenging to obtain using traditional in situ methods.

During the periods when soils and surface waters are not frozen, time series of streamflow can be partitioned into two primary components: 'event flow', which is a transient response to increased precipitation forcing; and 'baseflow', which represents the background or ambient drainage of the water stored in soils beneath the surface (Beven, 2001;Hall, 1968;Appleby, 1970;Horton, 1935). Streamflow recession analysis is a classical tool that has been used to investigate the ways in which

storage contributes to streamflow, and to derive information on storage properties and regional unconfined aquifer characteristics (Tallaksen, 1995;Rupp and Selker, 2005;Brutsaert, 2008;Rupp and Woods, 2008;Tague and Grant, 2004;Clark et al., 2009;Biswal and Marani, 2010;Shaw and Riha, 2012;Biswal and Nagesh Kumar, 2015). Brutsaert and Nieber (1977) first proposed plotting an observed recession slope of hydrograph to estimate the storage-discharge relationship. After decades of use in the hydrological sciences, this framework was expanded by Kirchner (2009) in the simple dynamical systems

approach, under the fundamental assumption that the discharge of the stream depends solely on the amount of water stored in the catchment. The motivation was to create a functional relationship between discharge and storage that could then be used to model discharge using only precipitation and evapotranspiration data. To date, there have been few studies on how low-flows or baseflow relate to total water storage (Krakauer and Temimi, 2011;Wittenberg and Sivapalan, 1999;Thomas et al., 2015;Wittenberg, 1999).

The relatively recent (e.g., 2000-current) availability of satellite-based Earth observations has generally improved our understanding of water stores and fluxes at varying scales, during normal and under extreme conditions (Alsdorf et al., 2010;Beighley et al., 2011;Swenson and Wahr, 2009;Kim et al., 2009;Reager et al., 2014;Sproles et al., 2015;Riegger and Tourian, 2014;Riegger, 2018;Tourian et al., 2018). For example, the Gravity Recovery and Climate Experiment (GRACE) satellites launched in 2002 provide monthly changes in total water storage resulting from water mass effect on the Earth's

gravity field (Tapley et al., 2004). These changes are computed as total terrestrial water storage anomalies (TWSA) and describe the monthly difference in storage state from the record-length mean. Because of the ability of the satellite to measure changes in the entire vertical column, including surface and subsurface water storage, these first-of-their-kind measurements have provided a valuable tool in understanding seasonal and interannual subsurface changes in water storage.

Building on these previous efforts and concepts, exponential relationships between monthly, non-winter discharge and GRACE

TWSAs are developed at 12 U.S. Geological Survey streamflow gauge locations distributed throughout the Mississippi River Basin (Fig. 1, Table 1) for a 12.5-year period (April 2002 to October 2014). A forward-looking, low-flow filter is applied to the sorted discharge-TWSA pairs as a baseflow proxy. Exponential relationships between discharge and TWSA are developed for all non-winter flows and approximated baseflows. Results are used to investigate the fraction of non-winter monthly discharge approximated as baseflow throughout the Mississippi River basin.

We define drainable water storage as "the volume of water in a basin that is connected to streamflow and would drain out of the basin as time went towards infinity with no additional precipitation inputs". Tourian et al. (2018) was the first study to estimate a total drainable water storage from a large river basin. This was done by estimating a linear relationship between the storage variability with the discharge at the mouth and applying a phase shift between the two time-series using a Hilbert transform. Here, to characterize the drainable storage from the sub-basins, GRACE TWSAs are transformed into drainable

water storages (i.e., not anomalies) using the derived discharge-TWSA relationships. Applying baseflow recession allows for non-linearity in the discharge-storage relationship by treating only the case of storage driven flow (baseflow). For the first time, we demonstrate the direct relationship between storage and discharge on a basin and sub-basin scale, we estimate parameters in the baseflow recession equation and we give the first estimate of a new quantity (drainable basin storage) that has never been estimated using only observations.

## 2 Data and Methods

### 2.1 Data

The GRACE data used here are the GRCTellus JPL RL05 Mass Concentration (mascon) solution data (Watkins et al., 2015;Wiese, 2015). This GRACE Total Water Storage Anomaly (TWSA) product is a 0.5-degree grid based on the spatial variability of the 3-degree measurements. The TWSA data for the Mississippi subbasins are aggregated over each subbasin using the area-weighted averaging method presented by Riegger and Tourian (2014). Due to satellite battery management and other issues, there are some missing months in the GRACE dataset. In total, 12 of the 151 monthly values are missing in our period of study. To fill missing months, linear interpolation between the previous and following months was used.

Monthly streamflow measurements ($Q_o$) were obtained for select discharge gauge stations (U.S. Geological Survey, 2015). The gauge stations were selected based on data availability, drainage area and location throughout the Mississippi basin (i.e., along major tributaries). The 12 sites were distributed throughout the Mississippi Basin with three along the Ohio River (1-3), three along the Upper Mississippi River (4-6), five along the Missouri River (7-11), and one near the outlet of the Mississippi River (12) (Fig. 1). Rodell and Famiglietti (1999) estimated that the minimum region size in which GRACE could resolve water mass variability would be about 200,000 km2, a smaller size than our smallest basin. The GRACE mascons (Watkins et al., 2015) are statistically independent and are at a 3-degree resolution (around 90,000 km2). Although multiple sites are from individual tributaries, they are distributed along the river such that the difference in drainage area between two sites is roughly 100,000 km$^2$ or more.

All relevant gauge information, such as river name, drainage area, and period of record, is contained in the Table 1. It is essential to note that potential cold weather months (November through March) were excluded from this analysis for USGS streamflow to minimize the impacts of snow and ice influence on the total water storage. For example, if basin-wide storage increases due to snow accumulation, it is likely that there will be no correlated change in discharge at that time. Thus, the storage change measured by GRACE for those months is not directly linked to discharge until some later period. The sensitivity of the results of this study to the selection of April through October as the non-frozen period is likely to be minimal in this region.

There are other possible sources of storage variability that should be considered when using GRACE measurements, such as vegetation growth and groundwater pumping. Regarding vegetation biomass, Rodell et al. (2007) affirms that the seasonal and interannual biomass variations are typically smaller than the uncertainty in the GRACE TWSA measurements, and based on

the global maps of vegetation biomass (Rodell et al., 2005), this holds true for the Mississippi River Basin. Significant pumping occurs in the High Plains located in the basin, however, being a shallow-water-table aquifer (Scanlon et al., 2012;Brookfield et al., 2018;Nie et al., 2018), the storage changes would still be linked to baseflow generation. In other words, the portions of the basin which are experiencing water table decline due to human activities would still exhibit the same general storage-
discharge relationship.

## 2.2 Methods

To identify potential relationships between monthly discharge ($Q$) and basin storage ($S$), GRACE TWSA data are used to represent storage variability and paired time series of $Q$-$S$ are determined for each sub-basin.  Mean monthly observed discharge ($m^3 s^{-1}$) is converted to depth units (cm $month^{-1}$) by cumulating flow rates for each month and dividing by the drainage
area upstream of each site (Table 1).  Only non-winter months were selected to limit the impacts of snow processes on $Q_o$-$S$ relationships. Following work by Kim et al. (2009), we focus on the fact that most summer storage variability in the Mississippi River basin is not due to surface water storage, but instead to sub-surface storage (including vadose zone). Our assumptions are applied to the recession of the streamflow records, namely that baseflow drives the portion of streamflow that underlies monthly peaks, and that this baseflow amount can be regressed against storage to achieve the storage minimum with calculated
uncertainty. Pairing $Q_o$ with $S$, we also assume that an average monthly discharge corresponds to the GRACE TWSA for the same month, which derives from a single measurement in the concerning month. However, the GRACE solution integrates temporal information from several ground tracks through the study region into the monthly gravity field, a single value carrying information of a whole month. Note that we focus on storage anomalies rather than absolute water storage to determine the discharge relationships because of the inability to quantify absolute storage based only on GRACE measurements.
To investigate baseflow ($Q_b$) relationships, a forward-looking 'low-flow filter' is developed and applied. The rationale for the filter is that there are both baseflow and event flow represented in the discharge record at any time, but only the baseflow portion of streamflow serves to infer drainable storage.  Hence, we assume that the storage-driven portion of discharge generally increases with increasing $S$, here represented by GRACE TWSA.  To build the $Q_b$-$S$ relationship, the $Q_o$-$S$ paired series is sorted from the minimum to maximum value of $S$. Because $Q_o$ is assumed to increase with $S$, $Q_b$ for a given $S$ is set
to the forward-looking minimum $Q_o$. Next, a $Q_b$ value is estimated for each $S$, based on minimum measured values of $Q_o$:

$$Q_b(S_i) = min \, |Q_o(S_i)|_{i=1}^{n}$$

where $n$ is the number of forward-looking values remaining in the paired series. In other words, the filter looks at the next n $Q_o$ values paired to the next n larger S values, selecting the minimum $Q_o$ as baseflow. The value of n can be subjective depending on the series size. Here, we used 20% of the number of pairs (18 months), after analyzing the model's sensitivity
to $n$ (Figure S1). The process defines the low-flow envelope in the $Q_o$-$S$ series, where the variations in discharge above the minimum value are due to short duration rainfall-runoff events not captured in the monthly GRACE TWSAs. Here, we term the low-flow series as baseflow ($Q_b$) but acknowledge our definition of baseflow may differ from other studies.

Building on previous studies (eg., Kirchner, 2009;Reager et al., 2014), which suggest that summer river discharge and drainable storage generally show an exponential relationship, we assume a relationship for total discharge and estimated baseflow in the form of Eq. (2):

$$Q = \alpha e^{\beta S}$$

where Q is the non-winter discharge ($Q_o$) or estimated baseflow ($Q_b$), $\alpha$ and $\beta$ are coefficients, and $S$ is basin storage defined here as GRACE TWSA.

To transform TWSA into an absolute water storage value, referenced herein as drainable storage ($S_e$) that directly influences discharge, a storage offset must best estimated. For example, Riegger and Tourian (2014) proposed a definition of time-dependent water absolute storage $S_e(t)$, using Eq. (3):

$$S_e(t) = TWSA(t) + S_o$$

where $TWSA(t)$ is the monthly storage anomaly and $S_o$ is an unknown constant storage offset. $S_o$ only shifts the $S_e(t)$ series without impacting its temporal variability. Figure 2 shows how the TWSA's provide the same fit (e.g., $R^2$) and exponential coefficient ($\beta$) accounting for the change in discharge with changing storage. Only the leading coefficient ($\alpha$) changes in response to the value of the storage offset ($S_o$) being added to each TWSA.  The intent of Figure 2 is to demonstrate that TWSA

and S can be used interchangeably by replacing $\alpha$ to account for the resulting desired storage units. The storage offset cannot be measured directly but should correspond to the long-term mean water storage for the region of interest. Based on the assumption that baseflow is driven by storage ($S_e$) and therefore a linear function of storage, the relationship between discharge and $TWSA$ can provide insights for estimating the representative $S_o$ value, which provides an opportunity to estimate drainable storage.

**3 Results and Discussion**

**3.1 Discharge-Storage Relationships**

As discussed, we assume there is an exponential relationship between storage and discharge. However, because we base our $Q$-$S$ relationship only on measurements, we use GRACE TWSA as a surrogate of storage. Figure 3 shows all non-winter (Apr-Oct) monthly observed discharges ($Q_o$) and the relationships between discharge and storage or all 12 sub-basins.  In general,

the figure shows that the Ohio and Upper Mississippi sub-basins (1-6) exhibit similar behavior in terms of magnitude and variability of discharge, while the Missouri sub-basins (7-11) have much less variability and smaller discharges for a given storage.  Note that, the variability observed in the Missouri sub-basins (7-11) series is due to high $Q$-$S$ points resulting from flooding in April to July 2011 (Reager et al., 2014), where the four largest storages are from these months. Figure 3 also shows how the $Q_b$-$S$ relationships capture the minimum flow conditions for the observed discharge-storage series (i.e., minimum flow

envelope). The variability above the $Q_b$-$S$ curve represents short-duration event discharges not captured by storage driven discharge.

The resulting $\alpha$, $\beta$ and $R^2$ values for the $Q_o$-$S$ and $Q_b$-$S$ relationships are shown in Figure 3 and listed in Table S1. In general, the relationships fit the $Q_b$-$S$ pairs with a median $R^2$ of 0.89 ranging from 0.46 to 0.92. For overall discharge, which includes event variability, the median $R^2$ drops to 0.63 ranging from 0.40 to 0.80. The $\alpha$ values range from 0.15 to 1.5 (cm month$^{-1}$) for baseflow and 0.22 and 2.7 (cm month$^{-1}$) for streamflow and differ between the major tributaries. In general, $\alpha$ tends to decrease as minimum observed discharge decreases. For example, values along the Missouri River are noticeably lower than those along the upper Mississippi and Ohio Rivers. As expected, both $\alpha_b$ and $\alpha_o$ are highly correlated with mean annual low-flow ($R$ is 0.99 for baseflow and 0.96 for streamflow).

Comparing the two relationships, $\alpha_b$ is equal to roughly 65% of $\alpha_o$ ranging from 52-75%. Note that, the ratio $\alpha_b/\alpha_o$ represents the mean baseflow fraction at each station when the TWSA is zero (i.e., $Q_b = \alpha_b$ and $Q_o = \alpha_o$), which corresponds to the mean storage observed during the GRACE period. Although baseflow fractions are difficult to assess and vary based on estimation methods (Cheng et al., 2016;Eckhardt, 2008;Gonzales et al., 2009;Lott and Stewart, 2016;Zhang et al., 2017), the values reported here are consistent with those in the literature. Zhang and Schilling (2006) reported ratios ranging from 65-75% for sites along the Mississippi River. Arnold et al. (2000) reported a ratio of 65% in the upper Mississippi River. Beighley et al. (2002) reported a median ratio of 55% for the Susquehanna River, which boarders the Ohio on its eastern boundary.

The $\beta$ values (i.e., exponential coefficient that scales discharge based on $S$) range from 0.02 to 0.1 for baseflow and 0.04 and 0.1 for streamflow and differ between the major tributaries. Based on a qualitative assessment, $\beta$ appears to decrease as the amount of water regulation increases. For example, the Missouri River is known to be highly regulated and the associated $\beta$ values are noticeably lower than those for the upper Mississippi and Ohio Rivers. In a regulated system, basin storage can increase with little change in river discharge because water is being stored in lakes/reservoirs. In this case, the Missouri river has several very large reservoirs (e.g., Lake Oahe, Lake Sakakawea, Fort Peck Lake), which may explain the relative lower relation between $Q$-$S$. This is one of this method's limitations, creating an uncertainty from the inability to include specific basin characteristics. For this reason, the relationships for heavy regulated rivers only reflect reservoir storage availability observed during the study period. Of interest is the difference in $\beta_o$ and $\beta_b$ along the Missouri River, where $\beta_b$ is roughly 35-62% of $\beta_o$ as compared to the other rivers where $\beta_b$ is 84-110% of $\beta_o$. This difference, which is due to disproportionally lower $\beta_o$ values for the Missouri River, suggests that in regulated systems storage changes are mitigated more for baseflow as compared to event-flow conditions (Fig. 3). As expected, the $\beta$ values are correlated with streamflow variability, defined here as the ratio of mean annual low-flow divided by mean annual flow for non-winter months ($Q_{m\text{-}min}/Q_m$), where $R$ is -0.89 and -0.94 for baseflow and streamflow, respectively. The correlation of $\alpha$ to low-flows and $\beta$ to streamflow variability supports the physical meaning of $Q$-$S$ relationships (Kirchner, 2009;Reager et al., 2014).

## 3.2 Absolute Water Storage

A unique aspect of the $Q$-$TWSA$ relationship described in equation 2 is that it can be used to estimate the storage offset ($S_o$) in equation 3, which enables the conversion of $TWSA$ to drainable storage. For example, solving equation 2 for $TWSA$ when streamflow is approximately zero, yields the maximum negative $TWSA$ for the associated $Q$-$TWSA$ relationship. If we set the

storage offset to the maximum negative *TWSA* in equation 3, we can convert *TWSA* to drainable storages, where the basin storage is zero for the near zero flow condition. This is the fundamental concept supporting the assumed *Q-S* relationships. The challenge is defining near zero streamflow because an exponential relationship cannot be solved for *S* if *Q* is zero. Here, we assume near zero streamflow is approximately 0.01% to 0.1% of the minimum monthly non-winter observed discharge (see $Q_{min}$ in Table 1). Although this is not exact, it is bounded by observed streamflow and provides discharges that capture the extreme hydrologic conditions associated with zero drainable storage. For example, 0.1% $Q_{min}$ corresponds to mean monthly discharges ranging from only 0.1 to 4.5 $m^3 s^{-1}$ between sites. Using the above approach and the $Q_o$-*TWSA* relationships in Fig. 3, Figure 4 shows the non-winter (Apr-Oct) drainable storage for each sub-basin during the study period, where the colored regions represent the range in storage measured by GRACE for the two estimates of storage offset (S$_o$ for 0.1% $Q_{min}$ and 0.01% $Q_{min}$).

Since the Mississippi River station (Site 12) resulting storage offset ranges from 96 to 123 cm (i.e. 109± 14 cm) and the observed basin-wide TWSA ranges -9.7 to 14.6 cm, we estimate the absolute drainable storage as 2,900 ± 400 to 3,600 ± 400 km³. Considering that the Mississippi River site drains all 11 sub-basins with sites 3, 6 and 11 representing the upper Mississippi, Ohio and Missouri river outlets (2.3 million km²). There is roughly 600,000 km² of drainage area above Site 12 not captured by three outlet gauges. Using the average storage per km² from the three sub-basins, we estimate storage for the remaining area. Cumulating the sub-basin and ungauged storages, we estimate that the Mississippi River Basin storage offset varies from 3,100 to 4,000 km³ for non-winter months (Site 12* in Fig 4), i.e. approximately one tenth of the maximum storage in the largest U.S. reservoir: Lake Mead. Although there should be no difference in the storage offset from the two approaches, a difference of roughly 10% is found, which may result from the storage per unit area from the sub-basins over-estimating the storage in the ungauged area. Although the range of mean storage is 800 to 900 km³, it represents less than 30% of the lowest storage estimates. Thus, we provide one of the first drainable storage estimates for the Mississippi River Basin and its major tributaries. These values cannot be validated since there are no current measurements of such an amount. Most large-scale models (e.g, PCR-GLOBWB, van Beek and Bierkens, 2009) are not fully coupled with groundwater models and contain structural errors on the ability to represent the GRACE-observed storage variability (Houborg et al., 2012;Scanlon et al., 2018). Thus, the comparison would not be direct. The storage offsets listed in Table S2 can be used to covert GRACE TWSA time series to absolute drainable storage time series and determine corresponding α values.

## 4 Conclusions

Given the importance of knowing how much water is available for societal demands and the complexity to measure this quantity with traditional methods, the primary goals of this research are to estimate total drainable water storage and the fraction of baseflow in the Mississippi River basin using remotely sensed measurements.

In summary, our approach focuses on non-winter months (Apr-Nov) for the period of April 2002 through October 2014 for 12 watersheds distributed throughout the Mississippi Basin. A forward-looking, low-flow filter is used to approximate baseflow

from measured discharges. Exponential relationships between discharge and NASA's GRACE total water storage anomalies are developed for all 12 sub-areas. The relationships show that the fraction of baseflow in the sub-basins varies from 52 to 75% regionally. The provided approach can be used to provide estimates of drainable water storage for watersheds larger than roughly 200,000 km$^2$ using only measurements derived the GRACE mission and monthly streamflow gauge observations.

Since we base our analysis on observed quantities, a certain level of empiricism is required to validate the methodology. Still, we believe that this analysis is an initial step towards further understanding the relationship between storage and discharge. Future research is recommended to: investigate the effects of temporal sub-sampling in developing $Q$-$S$ relationships; explore additional methods for estimating baseflow values for each increasing storage change value; explore additional methods to estimate $S_o$ with and/or without measured discharges; and integrate winter months into the analysis to characterize year-round

discharge-storage relationships. Our long-term goal is to estimate discharge (e.g., baseflow) without gauge measurements to characterize and model hydrologic and ecological cycles in regions with limited or no in-situ measurements.

## 5 Data Availability

The GRACE mascon solution data (Wiese, 2015) can be accessed at ftp://podaac-ftp.jpl.nasa.gov/allData/tellus/L3/mascon/RL05/JPL/non-CRI/netcdf and the monitored discharge data (U.S. Geological

Survey, 2015) is provided by the National Water Information System and can be accessed at https://waterdata.usgs.gov/nwis.

## 6 Author Contribution

All authors conceptualized the project. HEM and REB performed the analysis, investigation and validation. HEM prepared the manuscript with contributions from all co-authors.

## 7 Competing Interests

The authors declare that they have no conflict of interest.

## 8 Acknowledgments

This work was funded in part by NASA's GRACE Science Team (NNN13D505T and NNX12AJ95G), Terrestrial Hydrology (NNX12AQ36G, NNX14AD82G), and SWOT Science Team (NNX16AQ39G) Programs. Also supported by the Brazilian government through CAPES (Coordenacao de Aperfeicoamento de Pessoal de Nivel Superior) scholarship

(88888.076230/2013-00). A portion of this research was performed at the Jet Propulsion Laboratory, California Institute of Technology, under contract with NASA.

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

**Table 1. USGS gauge information and streamflow statistics: mean annual non-winter monthly discharge (Qm), mean annual minimum non-winter monthly discharge (Qm-min), and minimum non-winter monthly discharge (Qmin) observed during the period of study.**

| ID | USGS Station | River | Drainage Area, $km^2$ | Period of Record | $Q_m$, cm month$^{-1}$ | $Q_{m-min}$, cm month$^{-1}$ | $Q_{min}$, cm month$^{-1}$ |
|---|---|---|---|---|---|---|---|
| 1 | 03303280 | Ohio | 251,000 | 1975/10-2015/09 | 3.40 | 1.01 | 0.40 |
| 2 | 03399800 | Ohio | 373,000 | 1993/10-2014/09 | 3.29 | 0.90 | 0.40 |
| 3 | 03611500 | Ohio | 526,000 | 1928/04-2015/01 | 3.34 | 1.18 | 0.47 |
| 4 | 05420500 | Upper Miss. | 222,000 | 1873/06-2015/11 | 2.30 | 1.00 | 0.53 |
| 5 | 05474500 | Upper Miss. | 308,000 | 1878/01-2015/11 | 2.42 | 0.90 | 0.44 |
| 6 | 05587455 | Upper Miss | 444,000 | 1997/10-2013/09 | 2.57 | 1.06 | 0.46 |
| 7 | 06185500 | Missouri | 233,000 | 1941/07-2015/10 | 0.31 | 0.22 | 0.13 |
| 8 | 06342500 | Missouri | 483,000 | 1927/10-2015/09 | 0.35 | 0.23 | 0.17 |
| 9 | 06610000 | Missouri | 836,000 | 1928/09-2016/03 | 0.37 | 0.29 | 0.17 |
| 10 | 06813500 | Missouri | 1,075,000 | 1949/10-2016/03 | 0.36 | 0.27 | 0.17 |
| 11 | 06935965 | Missouri | 1,357,000 | 2000/04-2015/12 | 0.56 | 0.32 | 0.20 |
| 12 | 07374000 | Mississippi | 2,916,000 | 2004/03-2016/04 | 1.33 | 0.67 | 0.40 |


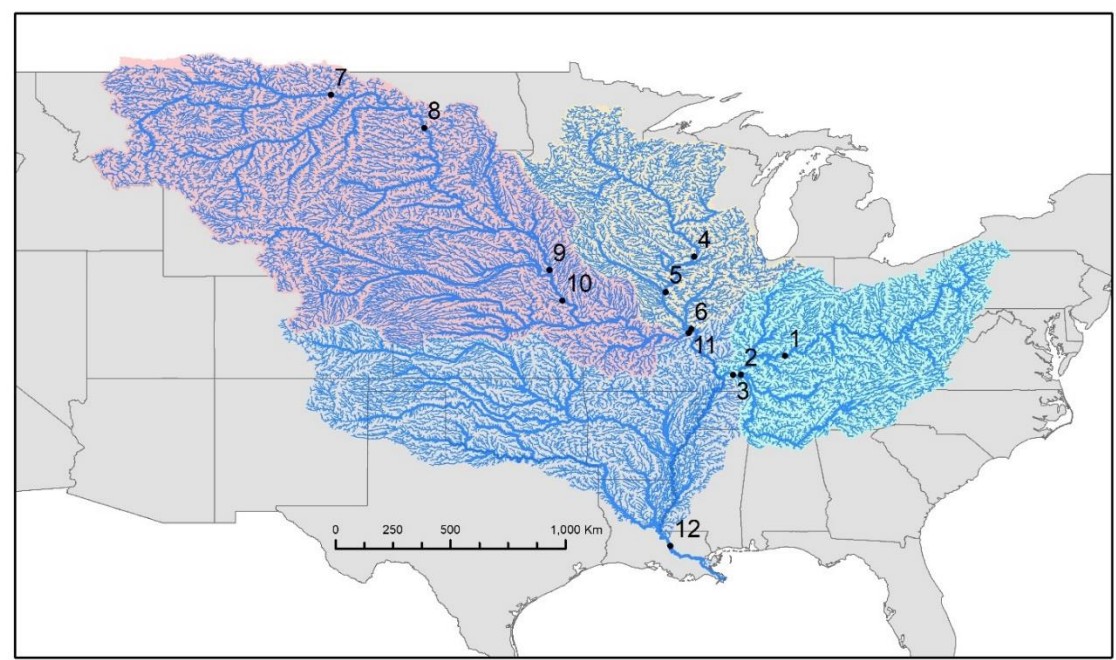

Figure 1: Study region with the location of selected USGS streamflow gauges.

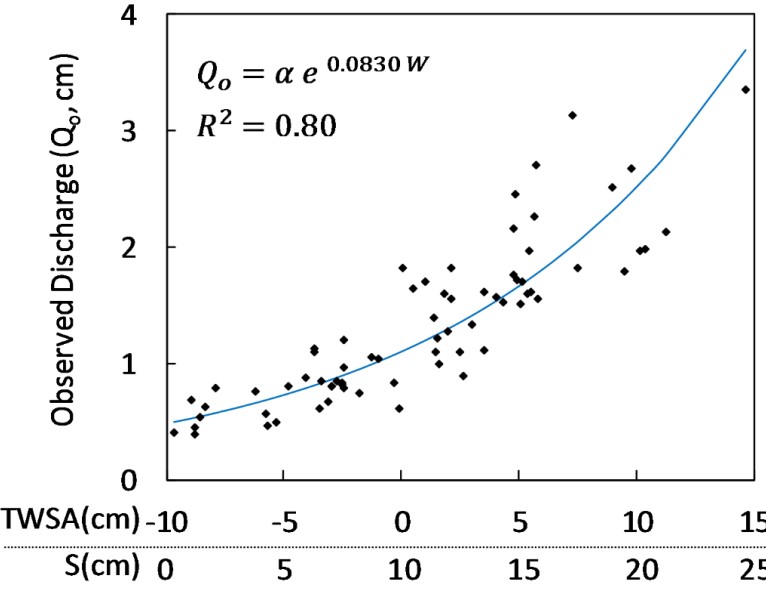

Figure 2: Storage-Discharge for the Mississippi River Basin (Site 12) based on Eq. (3) and an assumed $S_o$ value of 10 cm, which is arbitrarily selected to illustrate the effects on Q-S relationships, where $W$ represents storage in GRACE TWSA units (x-axis TWSA-cm) or absolute units (x-axis S-cm) and $\alpha$ is 1.101 if $W$ is TWSA or 0.4934 if $W$ is S.

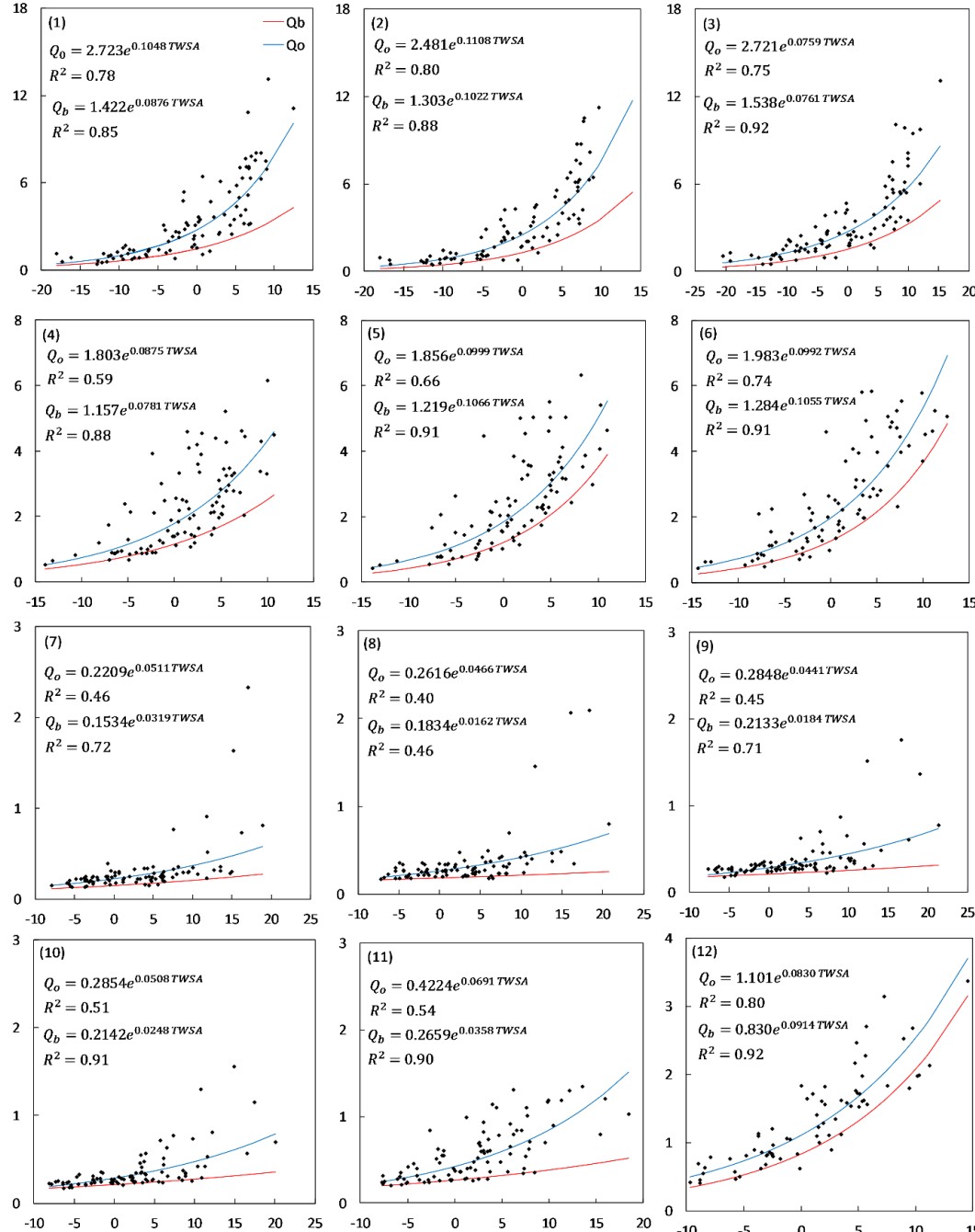

**Figure 3: Non-winter (Apr-Oct) monthly observed discharge ($Q_o$; y-axis in units of cm) and storage ($S$, x-axis in units of cm represented by TWSAs); the lines represent the relationship between observed discharge (blue) or baseflow (red) and storage. The plots IDs correspond to the site IDs listed Table 1 and shown in Figure 1. All relationships are significant at a 99% confidence interval (p-value < 0.00001), based on t-test.**

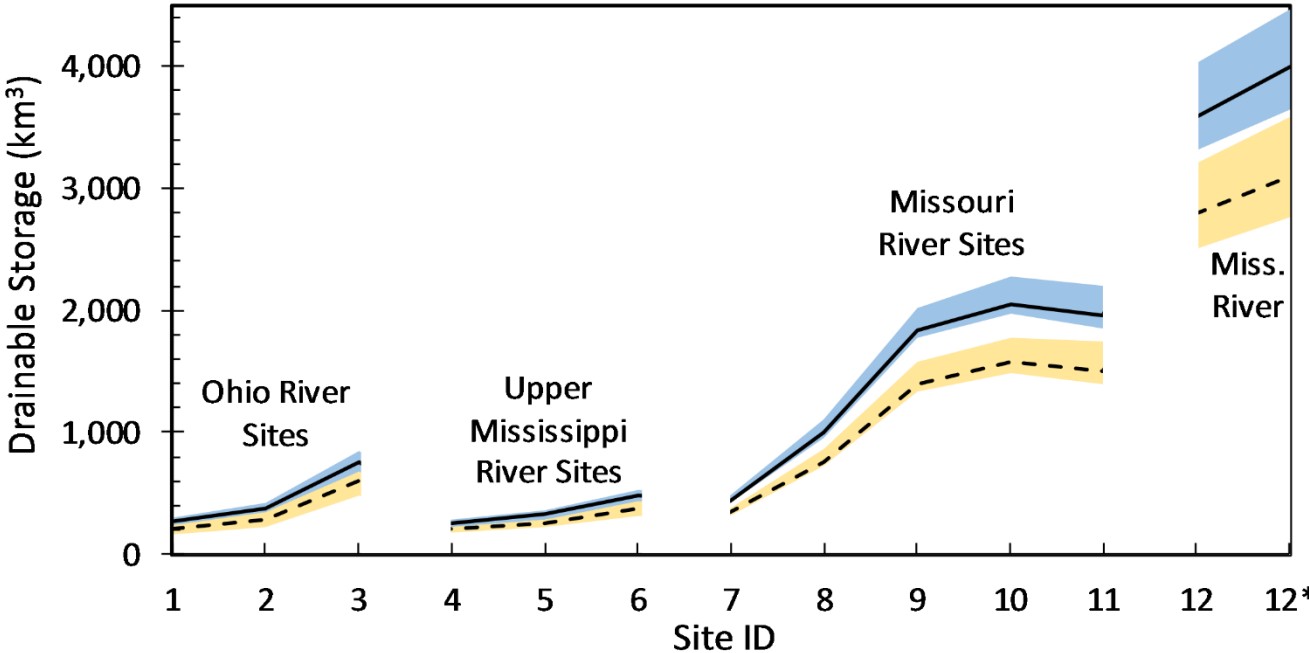

**Figure 4: Estimated drainable basin storages (S) for non-winter months (Apr-Oct) during the period 2002-2014 based on storage offsets derived using a zero-flow condition of 0.1% and 0.01% of Qmin; shaded regions show corresponding measured storage ranges from GRACE; sub-basin outlet locations are shown in Fig. 1; Site ID 12\* corresponds to estimated storage based on area-weighted values from Ohio, Upper Mississippi and Missouri River Basins.**