# Peer review of "Using GRACE in a streamflow recession to determine drainable water storage in the Mississippi River Basin"

_Hydrology and Earth System Sciences, 2019_

## Referee Comment (RC1) · Anonymous Referee #1 · 1 Apr 2019

General Comments This manuscript describes a method of relating streamflow measurements and terrestrial water storage anomalies (TWSA) from GRACE data products to estimate the drainable storage of several Mississippi River sub basins. This research is current, relevant, and of interest to the readers of HESS. The manuscript was well written and organized, and I enjoyed reading it. I have a couple of concerns with the fundamental concepts that underpin this research that require further explanation from the authors, as described under 'specific comments' below. In addition, I have further minor/editorial comments provided under 'technical corrections' below. Overall, I think this manuscript should be returned to the authors for major revisions.

[Figure]

Specific Comments First of all, the methodology estimating Qb is not clear. The authors state that the Qo-S pairs are ordered by size of S, then Qb is the 'forward-looking minimum' of Qo. Is this forward in time, or just in this ordered pairing from low to high S? I assume forward in time, because you can't simply ignore the order of events (a low S cannot be the result of a low Qo that won't occur for several months). In addition, the text says that Qb is estimated as a fraction of Qo using equation 1, yet equation 1 contains no metric for this fraction. Either this is the incorrect equation, or the term 'fraction' is used in error. The second concern is related to the temporal resolution of the data with respect to equation 1. GRACE data represent the TWSA on the particular day(s) the measurements were taken, and not the monthly high/low/average. Qo is defined as the mean monthly observed discharge. Thus, equation 1 is dependent upon pairing of an instantaneous value with a mean value. While some work indicates that TWSA variability is largely not due to surface water storage (storage that can fluctuate greatly with time), some evaluation of the variability of Q throughout each month should be considered before applying equation 1. Thirdly, while only considering non-winter storage variability simplifies the analysis with respect to snow accumulation and events, it does complicate the issue with respect to vegetation growth. The Mississippi basin is a large agricultural area, and a change in mass due to the increase in vegetation over the growing season should be addressed in this work. Along similar lines, I would be interested to know how much groundwater pumping takes place within each sub basin, and if that contributes significantly to changes in TWSA. Finally, while the authors address the issue of reservoir storage and releases and their influence over Q I think further work is needed to discuss how the Q-S relationships can still hold in these environments. If the flow of the stream is dependent upon reservoir releases they would not necessarily reflect the basin's storage (e.g. we can have a large reservoir release when groundwater levels (a reflection of baseflow) and drainable storage, are low), so how can the Q-S relationship still hold? Many reservoirs in the Mississippi basin are driven by downstream user demands and are not a reflection of what the natural flow conditions would be.

Technical Comments P3 L2: You provide an estimate of drainable basin storage, not total basin storage. P3 L16: 'smaller size' not 'inferior size' P3-4, L31-1: This sentence is redundant P4 L2: Recent research supports the conclusion that TWSAs are not due to surface anomalies, but also indicates that TWSAs are not related to water availability (drainage) in basins within the Mississippi. Areas with large vadose zones can have changes in the vadose zones dominate the changes in TWSA. P4 L5: Perhaps indicate that you focus on storage anomalies because it is not possible to quantify absolute storage with GRACE data. P5 L4-5: This should not be a surprise since you derived S from TWSA. P7 L6-11: This is a summary not conclusion. The conclusions need to be bolstered, at the moment they are quite weak. P7 L13: You didn't just use TWSA, you used Q as well. Figure 3: Are these regressions significant? Include axis labels. Figure 4 (and within text): This insinuates that drainable storage didn't change with time. How do you justify this in such a dynamic basin?

---

## Short Comment (SC1) · 3 Apr 2019

**General comments**

This manuscript describes a method of relating streamflow measurements and terrestrial water storage anomalies (TWSA) from GRACE data products to estimate the drainable storage of several Mississippi River sub basins. This research is current, relevant, and of interest to the readers of HESS. The manuscript was well written and organized, and I enjoyed reading it. I have a couple of concerns with the fundamental concepts that underpin this research that require further explanation from the authors, as described under 'specific comments' below. In addition, I have further minor/editorial

comments provided under 'technical corrections' below. Overall, I think this manuscript should be returned to the authors for major revisions.

*Response: We thank the reviewer for the support and the feedback to improve our manuscript. Detailed responses to your concerns are provided below, along with suggestions for changes in the manuscript for a posterior resubmission.*

**Specific comments**

First of all, the methodology estimating Qb is not clear. The authors state that the Qo-S pairs are ordered by size of S, then Qb is the 'forward-looking minimum' of Qo. Is this forward in time, or just in this ordered pairing from low to high S? I assume forward in time, because you can't simply ignore the order of events (a low S cannot be the result of a low Qo that won't occur for several months). In addition, the text says that Qb is estimated as a fraction of Qo using equation 1, yet equation 1 contains no metric for this fraction. Either this is the incorrect equation, or the term 'fraction' is used in error.

*Response: We understand your concern, and the lack of clarity on the estimation method for Qb. To reiterate, the equation we apply assumes that 'baseflow' comprises the storage-driven portion of the streamflow, but that there are other portions of streamflow contributed by surface runoff generation (i.e. not 'baseflow'). The discharge at any time is some combination of those two processes, and while the baseflow varies depending only on storage, the surface runoff varies based on other processes including precipitation rate, land cover type, and surface soil saturation, and tends to vary more rapidly.*

*After pairing GRACE TWSA with observed monthly discharge for each basin, the paired time series is sorted from minimum to maximum value of S. Then, for each pair, the filter looks at the next 18 discharge values, selecting the minimum value as the baseflow. Because the storage driven baseflow represents a partial component of the total streamflow corresponding to the minimum surface runoff (or ideally zero surface runoff) situation, we aim to find the case when the non-baseflow signal is min-*

*imized and baseflow dominates. The TWSA-discharge pair doesn't change, but this method selects a minimum discharge value that was measured (realistic) for a similar size (in the same magnitude bin) of storage, in hopes that this will best represent only the baseflow portion of the discharge signal.*

*You are correct about the use of the word "fraction". We will modify P4-L12 to read: "Next, a Qb value is estimated for each S, based on minimum measured values of Qo:"*

The second concern is related to the temporal resolution of the data with respect to equation 1. GRACE data represent the TWSA on the particular day(s) the measurements were taken, and not the monthly high/low/average. Qo is defined as the mean monthly observed discharge. Thus, equation 1 is dependent upon pairing of an instantaneous value with a mean value. While some work indicates that TWSA variability is largely not due to surface water storage (storage that can fluctuate greatly with time), some evaluation of the variability of Q throughout each month should be considered before applying equation 1.

*Response: The reviewer is correct in that we assume that an average discharge value for the whole month is analogous to, or corresponds to, the GRACE monthly solution. In fact, a monthly GRACE solution may integrate temporally information from several ground tracks through the study region into the monthly gravity field, and each of those ground tracks could have been recorded on a separate day of that month. So, it is also erroneous to say that a monthly GRACE solution represents a single day over the study region. If the study region is relatively large, like the river basins here, it is highly likely that several samples of information throughout the month are included in the solution. With this in mind, the issue the reviewer has identified is worth mentioning, but there is no clear path to overcome this issue at this time. It is not clear how the daily analysis of discharge would offer any insights, in terms of the fraction of discharge that is driver by baseflow or surface water, and an (e.g.) statistical analysis of the discharge time series alone with no complimentary information from GRACE would not really offer a new methodological approach.*

*As the reviewer mentioned, following work by Kim et al. (2009) to partition variability in the GRACE signal in global river basins, we focus on the fact that most summer storage variability in the Mississippi River basin is primarily not due to surface water storage, but instead to sub-surface storage in soils, and therefore lend themselves well to a baseflow recession analysis.*

*As such, we should mention this point as a caveat of the study, and text to this effect will appear in the methods section.*

Thirdly, while only considering nonwinter storage variability simplifies the analysis with respect to snow accumulation and events, it does complicate the issue with respect to vegetation growth. The Mississippi basin is a large agricultural area, and a change in mass due to the increase in vegetation over the growing season should be addressed in this work. Along similar lines, I would be interested to know how much groundwater pumping takes place within each sub basin, and if that contributes significantly to changes in TWSA.

*Response: Our study was focused on non-winter storage variability by necessity: this approach provides the best look at the storage-discharge relationship without the complication of freeze-thaw processes on both storage and discharge. Based on global maps of vegetation biomass (Rodell et al., 2005), Rodell et al. (2007) affirms that the seasonal and interannual biomass variations are typically smaller than the uncertainty in the GRACE TWSA measurements. While still a source of uncertainty that should be cited in our work (and will be fixed in the final manuscript), this also holds true for the Mississippi River basin.*

*Groundwater pumping is really a separate topic. Significant pumping does occur in the High Plains aquifer, which is a shallow-water-table aquifer. As such, in the case for which changes in storage from groundwater pumping would lower the water table, those storage changes would still be linked directly to baseflow generation. So those storage changes would be generally consistent with the current approach and hypothe-*

*sis. In other words, the portions of the basin which are experiencing water table decline due to human activities would still exhibit the same general storage-discharge relationship, and while an in-depth analysis of groundwater pumping activities would theoretically interesting, it would not augment the results of our study, nor provide coherent insight on our results. There are already several studies on the High Plains aquifer using GRACE to monitor groundwater changes (e.g. Scanlon et al., 2012;Brookfield et al., 2018;Nie et al., 2018).*

Finally, while the authors address the issue of reservoir storage and releases and their influence over Q I think further work is needed to discuss how the Q-S relationships can still hold in these environments. If the flow of the stream is dependent upon reservoir releases they would not necessarily reflect the basin's storage (e.g. we can have a large reservoir release when groundwater levels (a reflection of baseflow) and drainable storage, are low), so how can the Q-S relationship still hold? Many reservoirs in the Mississippi basin are driven by downstream user demands and are not a reflection of what the natural flow conditions would be.

*Response: To say that the streamflow is dependent upon reservoir releases is true to some extent for the smaller tributaries in the study domain and we should do a better job of clarifying our assumptions. For the larger river basins and their major rivers, streamflow shows a first-order response to precipitation and storage changes within the basin. The higher order "errors" introduced in our approach due to the misrepresentation of natural discharge would affect our recession approach, but there are challenges in quantifying these errors directly. Considering the timescales of a rain storm and runoff event, we assume that most reservoir operations would only significantly affect the downstream (i.e. large river) discharge due to small reservoir operations within a finite time-span and with finite storage volume (i.e. approximately 5-10*

*We discuss the effect of heavy regulation in the Missouri River Q-S relationships at P5 L31- P6 L6 and will add more text on this topic. This is one of the method's limitations, creating an uncertainty from the inability to include specific basin characteristics. This*

*effect is more pronounced for some of the sub-basins (Missouri River) than it is for others reflecting on a relative lower relation between Q-S. This is clear from the R2 in the panels 7-9 at Figure 3. However, we should make it clear that we understand that this is a limitation in the text as well.*

**Technical comments**

1)P3 L2: You provide an estimate of drainable basin storage, not total basin storage.

2)P3 L16: 'smaller size' not 'inferior size'

3)P3-4, L31-1: This sentence is redundant

5)P4 L5: Perhaps indicate that you focus on storage anomalies because it is not possible to quantify absolute storage with GRACE data.

6)P5 L4-5: This should not be a surprise since you derived S from TWSA.

*Response: Comments 1, 2, 5 are very pertinent, they should be easily incorporate into our text.*

*Comments 3, 6: We understand how these sentences can be seen as redundant, we included them for clarity.*

4)P4 L2: Recent research supports the conclusion that TWSAs are not due to surface anomalies, but also indicates that TWSAs are not related to water availability (drainage) in basins within the Mississippi. Areas with large vadose zones can have changes in the vadose zones dominate the changes in TWSA.

*Response: We will add a statement to clarify that most of the variability in TWS in this region comes from soil water storage changes, including the vadose zone. However, we did not understand the reviewer comment about "water availability" not being related to TWSA."*

7)P7 L6-11: This is a summary not conclusion. The conclusions need to be bolstered,

at the moment they are quite weak.

*Response: We will change the text to clarify that this statement is a summary, not a conclusion. To that end, we will add the phrase "in summary". We will work on our conclusions for the next submission.*

8)P7 L13: You didn't just use TWSA, you used Q as well.

*Response: That's true – the method offers an approach based on coupled TWSA and Q measurements. We will modify the text to be more accurate.*

9)Figure 3: Are these regressions significant? Include axis labels.

*Response: All relationships are significant at a 99% confidence interval (p-value < 0.00001), based on t-test. The axis labels are described in the figure caption to avoid redundancy in the figure.*

10)Figure 4 (and within text): This insinuates that drainable storage didn't change with time. How do you justify this in such a dynamic basin?

*Response: Drainable storage relates to the long-term mean storage capacity of the basin. It is time-invariant by definition. While this may evolve in the long-term due to geological or land use changes, we offer a first estimate over the years 2002-2015.*

**References**

*Brookfield, A. E., Hill, M. C., Rodell, M., Loomis, B. D., Stotler, R. L., Porter, M. E., and Bohling, G. C.: In Situ and GRACE-Based Groundwater Observations: Similarities, Discrepancies, and Evaluation in the High Plains Aquifer in Kansas, Water Resources Research, 54, 8034-8044, doi: 10.1029/2018wr023836, 2018.*

*David, C. H., Famiglietti, J. S., Yang, Z.-L., and Eijkhout, V.: Enhanced fixed-size parallel speedup with the Muskingum method using a trans-boundary approach and a large subbasins approximation, Water Resources Research, 51, 7547-7571, doi: 10.1002/2014wr016650, 2015.*

Kim, H., Yeh, P. J. F., Oki, T., and Kanae, S.: Role of rivers in the seasonal variations of terrestrial water storage over global basins, Geophysical Research Letters, 36, doi: 10.1029/2009GL039006, 2009.

Nie, N., Zhang, W., Chen, H., and Guo, H.: A Global Hydrological Drought Index Dataset Based on Gravity Recovery and Climate Experiment (GRACE) Data, Water Resources Management, 32, 1275-1290, doi: 10.1007/s11269-017-1869-1, 2018.

Rodell, M., Chao, B. F., Au, A. Y., Kimball, J. S., and McDonald, K. C.: Global Biomass Variation and Its Geodynamic Effects: 1982–98, Earth Interactions, 9, 1-19, doi: 10.1175/ei126.1, 2005.

Rodell, M., Chen, J., Kato, H., Famiglietti, J. S., Nigro, J., and Wilson, C. R.: Estimating groundwater storage changes in the Mississippi River basin (USA) using GRACE, Hydrogeology Journal, 15, 159-166, doi: 10.1007/s10040-006-0103-7, 2007.

Scanlon, B. R., Faunt, C. C., Longuevergne, L., Reedy, R. C., Alley, W. M., McGuire, V. L., and McMahon, P. B.: Groundwater depletion and sustainability of irrigation in the US High Plains and Central Valley, Proceedings of the National Academy of Sciences, 109, 9320-9325, doi: 10.1073/pnas.1200311109, 2012.

---

## Referee Comment (RC2) · Anonymous Referee #2 · 10 Apr 2019

The paper aims to estimate the amount of drainable water storage in a basin using GRACE satellite and streamflow data. They develop a forward-looking, low flow filter to isolate base flow; while transforming GRACE based storage anomalies to provide estimates of absolute drainable water storage in the Mississippi River Basin. The work is of interest and suitable for this journal as it deals with a fundamental aspect of hydrology, and provides useful technique to investigate storage-outflow relationships of large watersheds. Overall, the paper is written well and the figures are clear. The paper, however, would benefit from some major revisions, especially with regards to the introduction and methods section. For this reason, I suggest the editor consider the revisions suggested below prior to making a decision on this manuscript.

[Figure]

Major comments:

- The authors reference other studies that have used remote sensing to estimate water storage in basin; after looking at the titles of those journal articles, it seems that at least 2 of those studies (Tourian et.al., 2018; Riegger, 2018) have attempted to estimate total drainable water storage in a basin using GRACE data. How are the methods used in the present study different from those analysis? If the methods are different, then why was a different method developed? If there is a significant overlap in methods, then what is the novel contribution of this study? The answers to these questions should be clearly integrated into the introduction, as the original contributions of the authors seem unclear. - As pointed out by referee#1, the methods section needs to be written better especially with regards to how Qb was estimated. It seems unclear as to which "20% of the number of pairs (months)" were used to get the minimum value. Also, it would be useful to include a figure that shows the sensitivity of the model to n in the supplementary document to solidify that 20% was indeed a correct forward looking limit. - The justification of using Q-S relationship in a highly regulated systems (like the Missouri River) needs to be added. Can the storage values obtained in these systems still be considered as the total drainable water storage? How do the reservoir operational policies affect the low flow values obtained? It might be useful to go deeper into one of these regulated systems to explain why the estimates obtained are still useful/valid there. - The authors claim that the total drainable storage volumes they obtain cannot be validated. Can large-scale hydrological models like PCR-GLOBWB be used to obtain similar values? There should be some acknowledgement of the ability or inability of large-scale hydrological models to estimate a similar value. - The conclusions section currently seems to be a summary of the methods used in the study and the scope of future work. This section should be expanded further to include some of the results obtained, as well as a discussion of why/where it is important to know the total drainable storage of a basin.

Minor comments:
P1 L24-26: The sentence does not read correctly. I suggest having a separate sentence to describe/summarize the remote sensing that has contributed to estimating watershed storage. P2 L11: "the desire" seems redundant. Suggestion: "The motivation was to create a functional relationship. ...." Figure 1: It would be useful to include the sub-basin boundaries on the map to help orient the readers P4 L25-34: While it is implied that the authors use this expression to estimate the absolute water storage, it might be useful to explicitly state that here. P5 L3-7: It would be more useful to integrate this paragraph into the methods section as there seems to be no results here. P5 L24: Replace with "which corresponds to the mean" P7 L2: Replace with "of such an amount"

References: Riegger, J., Quantification of Drainable Water Storage Volumes in Catchments and in River Networks on Global Scales using the GRACE and/or River Runoff, Hydrol. Earth Syst. Sci. Discuss., 2018, 1-27, doi: 10.5194/hess-2018-38, 2018. Tourian, M. J., Reager, J. T., and Sneeuw, N.: The Total Drainable Water Storage of the Amazon River Basin: A First Estimate Using GRACE, Water Resources Research, 54, 3290-3312, doi: 10.1029/2017WR021674, 2018.

---

## Author Comment (AC1) · 10 May 2019

**Author's comments – Referee 1**

**General Comments:** This manuscript describes a method of relating streamflow measurements and terrestrial water storage anomalies (TWSA) from GRACE data products to estimate the drainable storage of several Mississippi River sub basins. This research is current, relevant, and of interest to the readers of HESS. The manuscript was well written and organized, and I enjoyed reading it. I have a couple of concerns with the fundamental concepts that underpin this research that require further explanation from the authors, as described under 'specific comments' below. In addition, I have further

minor/editorial comments provided under 'technical corrections' below. Overall, I think this manuscript should be returned to the authors for major revisions.

*Response: We thank the reviewer for the support and the feedback to improve our manuscript. Detailed responses to your concerns are provided below, along with suggestions for changes in the manuscript for a posterior resubmission.*

**Specific Comments:**

**Comment 1:** First of all, the methodology estimating Qb is not clear. The authors state that the Qo-S pairs are ordered by size of S, then Qb is the 'forward-looking minimum' of Qo. Is this forward in time, or just in this ordered pairing from low to high S? I assume forward in time, because you can't simply ignore the order of events (a low S cannot be the result of a low Qo that won't occur for several months). In addition, the text says that Qb is estimated as a fraction of Qo using equation 1, yet equation 1 contains no metric for this fraction. Either this is the incorrect equation, or the term 'fraction' is used in error.

*Response: We understand your concern, and the lack of clarity on the estimation method for Qb. To reiterate, the equation we apply assumes that 'baseflow' comprises the storage-driven portion of the streamflow, but that there are other portions of streamflow contributed by surface runoff generation (i.e. not 'baseflow'). The discharge at any time is some combination of those two processes, and while the baseflow varies depending only on storage, the surface runoff varies based on other processes including precipitation rate, land cover type, and surface soil saturation, and tends to vary more rapidly. After pairing GRACE TWSA with observed monthly discharge for each basin, the paired time series is sorted from minimum to maximum value of S. We are not ignoring the order of events, because even after sorting, the pairs still correspond to the same month. Then, for each pair, the filter looks at the next 18 discharge values (next 18 Qo paired to the next 18 larger S values), selecting the minimum value as the baseflow. Because the storage driven baseflow represents a partial component of the*

*total streamflow corresponding to the minimum surface runoff (or ideally zero surface runoff) situation, we aim to find the case when the non-baseflow signal is minimized and baseflow dominates. The TWSA-discharge pair doesn't change, but this method selects a minimum discharge value that was measured (realistic) for a similar magnitude (in the same magnitude bin) of storage, in hopes that this will best represent only the baseflow portion of the discharge signal. You are correct about the use of the word "fraction".*

***Text will be modified at P4 L10-15:*** *To build the Qb-S relationship, the Qo-S fixed paired series is sorted from the minimum to maximum value of S. Because Qo is assumed to increase with S, Qb for a given S is set to the forward-looking minimum Qo. Next, a Qb value is estimated for each S, based on minimum measured values of Qo: (equation 1) where n is the number of forward-looking values remaining in the paired series. In other words, the filter looks at the next n Qo values paired to the next n larger S values, selecting the minimum Qo as baseflow. The n value can be subjective depending on the series size. Here, we used 20% of the number of pairs (18 months), after analyzing the model's sensitivity to n.*

**Comment 2:** The second concern is related to the temporal resolution of the data with respect to equation 1. GRACE data represent the TWSA on the particular day(s) the measurements were taken, and not the monthly high/low/average. Qo is defined as the mean monthly observed discharge. Thus, equation 1 is dependent upon pairing of an instantaneous value with a mean value. While some work indicates that TWSA variability is largely not due to surface water storage (storage that can fluctuate greatly with time), some evaluation of the variability of Q throughout each month should be considered before applying equation 1.

***Response:*** *The reviewer is correct in that we assume that an average discharge value for the whole month is analogous to, or corresponds to, the GRACE monthly solution. In fact, a monthly GRACE solution may integrate temporal information from several ground tracks through the study region into the monthly gravity field, and each of those*

*ground tracks could have been recorded on a separate day of that month. So, it is also erroneous to say that a monthly GRACE solution represents a single day over the study region. If the study region is relatively large, like the river basins here, it is highly likely that several samples of information throughout the month are included in the solution. With this in mind, the issue the reviewer has identified is worth mentioning, but there is no clear path to overcome this issue at this time. It is not clear how the daily analysis of discharge would offer any insights, in terms of the fraction of discharge that is driver by baseflow or surface water, and an (e.g.) statistical analysis of the discharge time series alone with no complimentary information from GRACE would not really offer a new methodological approach.*

*As the reviewer mentioned, following work by Kim et al. (2009) to partition variability in the GRACE signal in global river basins, we focus on the fact that most summer storage variability in the Mississippi River basin is primarily not due to surface water storage, but instead to sub-surface storage in soils, and therefore lend themselves well to a baseflow recession analysis. As such, we should mention this point as a caveat of the study, and text to this effect will appear in the methods section.*

***Text will be modified at P4 L1-6:*** *Following work by Kim et al. (2009), we focus on the fact that most summer storage variability in the Mississippi River basin is not due to surface water storage, but instead to sub-surface storage (including vadose zone). Our assumptions are applied to the recession of the streamflow records, namely that baseflow drives the portion of streamflow that underlies monthly peaks, and that this baseflow amount can be regressed against storage to achieve the storage minimum with calculated uncertainty. Pairing $Q_o$ with $S$, we also assume that an average monthly discharge corresponds to the GRACE TWSA for the same month, which derives from a single measurement in the concerning month. However, the GRACE solution integrates temporal information from several ground tracks through the study region into the monthly gravity field, a single value carrying information about carrying information of a whole month. Note that we focus on storage anomalies (i.e. GRACE TWSA) rather*

*than absolute water storage to determine discharge relationships.*

**Comment 3:** Thirdly, while only considering nonwinter storage variability simplifies the analysis with respect to snow accumulation and events, it does complicate the issue with respect to vegetation growth. The Mississippi basin is a large agricultural area, and a change in mass due to the increase in vegetation over the growing season should be addressed in this work. Along similar lines, I would be interested to know how much groundwater pumping takes place within each sub basin, and if that contributes significantly to changes in TWSA.

*Response: Our study was focused on non-winter storage variability by necessity: this approach provides the best look at the storage-discharge relationship without the complication of freeze-thaw processes on both storage and discharge. Based on global maps of vegetation biomass (Rodell et al., 2005), Rodell et al. (2007) affirms that the seasonal and interannual biomass variations are typically smaller than the uncertainty in the GRACE TWSA measurements. While still a source of uncertainty that should be cited in our work (and will be fixed in the final manuscript), this also holds true for the Mississippi River basin.*

*Groundwater pumping is really a separate topic. Significant pumping does occur in the High Plains aquifer, which is a shallow-water-table aquifer. As such, in the case for which changes in storage from groundwater pumping would lower the water table, those storage changes would still be linked directly to baseflow generation. So those storage changes would be generally consistent with the current approach and hypothesis. In other words, the portions of the basin which are experiencing water table decline due to human activities would still exhibit the same general storage-discharge relationship, and while an in-depth analysis of groundwater pumping activities would theoretically interesting, it would not augment the results of our study, nor provide coherent insight on our results. There are already several studies on the High Plains aquifer using GRACE to monitor groundwater changes (e.g. Scanlon et al., 2012;Brookfield et al., 2018;Nie et al., 2018).*
***Text will be added after at P3 L24-26:*** *There are other possible sources of storage variability that should be considered when using GRACE measurements, such as vegetation growth and groundwater pumping. Regarding vegetation biomass, Rodell et al. (2007) affirms that the seasonal and interannual biomass variations are typically smaller than the uncertainty in the GRACE TWSA measurements, and based on the global maps of vegetation biomass (Rodell et al., 2005), this holds true for the Mississippi River Basin. Significant pumping occurs in the High Plains aquifer located in the basin, however, being a shallow-water-table aquifer (Scanlon et al., 2012;Brookfield et al., 2018;Nie et al., 2018), the storage changes would still be linked to baseflow generation. In other words, the portions of the basin which are experiencing water table decline due to human activities would still exhibit the same general storage-discharge relationship.*

**Comment 4:** Finally, while the authors address the issue of reservoir storage and releases and their influence over Q I think further work is needed to discuss how the Q-S relationships can still hold in these environments. If the flow of the stream is dependent upon reservoir releases they would not necessarily reflect the basin's storage (e.g. we can have a large reservoir release when groundwater levels (a reflection of baseflow) and drainable storage, are low), so how can the Q-S relationship still hold? Many reservoirs in the Mississippi basin are driven by downstream user demands and are not a reflection of what the natural flow conditions would be.

***Response:*** *To say that the streamflow is dependent upon reservoir releases is true to some extent for the smaller tributaries in the study domain and we should do a better job of clarifying our assumptions. For the larger river basins and their major rivers, streamflow shows a first-order response to precipitation and storage changes within the basin. The higher order "errors" introduced in our approach due to the misrepresentation of natural discharge would affect our recession approach, but there are challenges in quantifying these errors directly. Considering the timescales of a rain storm and runoff event, we assume that most reservoir operations would only significantly*

*affect the downstream (i.e. large river) discharge due to small reservoir operations within a finite time-span and with finite storage volume (i.e. approximately 5-10% of the discharge signal). Studies on numerical modeling of the Mississippi river and the estimated effect of diversions and reservoirs at the gage support these estimates (e.g. David et al., 2015).*

*However, the same study by David et al. (2015) points that heavy regulation causes longer residence times and dampening of the flow, and the lack of representation of storage processes in reservoirs caused poor simulation of modeled flow in the Missouri River basin. This is one of this method's limitations, creating an uncertainty from the inability to include specific basin characteristics. This effect is a lot more pronounced for the Missouri River sub-basins than it is for others reflecting on a relative lower relation between Q-S. This effect is evidenced in the R2 in the panels 7-9 at Figure 3."*

**Text will be modified at P5-6 L31-6:** *Based on a qualitative assessment, $\beta$ appears to decrease as the amount of water regulation increases. For example, the Missouri River is known to be highly regulated and the associated $\beta$ values are noticeably lower than those for the upper Mississippi and Ohio Rivers. In a regulated system, basin storage can increase with little change in river discharge because water is being stored in lakes/reservoirs. In this case, the Missouri river has several very large reservoirs (e.g., Lake Oahe, Lake Sakakawea, Fort Peck Lake), which may explain the relative lower relation between Q-S (panels 7-9 at Fig. 3). This is one of this method's limitations, creating an uncertainty from the inability to include specific basin characteristics. For this reason, the relationships for heavy regulated rivers only reflects reservoir storage availability observed during the study period. Of interest is the difference in $\beta o$ and $\beta b$ along the Missouri River, where $\beta b$ is roughly 35-62% of $\beta o$ as compared to the other rivers where $\beta b$ is 84-110% of $\beta o$. This difference, which is due to disproportionally lower $\beta o$ values for the Missouri River, suggests that in regulated systems storage changes are mitigated more for baseflow as compared to event-flow conditions (Fig. 3).*

**Technical Comments**

**Comment 1)P3 L2:** You provide an estimate of drainable basin storage, not total basin storage.

**Comment 2)P3 L16:** 'smaller size' not 'inferior size'

**Comment 3)P3-4, L31-1:** This sentence is redundant

**Comment 5)P4 L5:** Perhaps indicate that you focus on storage anomalies because it is not possible to quantify absolute storage with GRACE data.

**Comment 6)P5 L4-5:** This should not be a surprise since you derived S from TWSA.

*Response: Comments 1, 2, 5 are very pertinent, they should be easily incorporate into our text.*

*Comments 3, 6: We understand how these sentences can be seen as redundant, we initially included them for enhanced clarity and could remove them in our resubmission.*

**Comment 4)P4 L2:** Recent research supports the conclusion that TWSAs are not due to surface anomalies, but also indicates that TWSAs are not related to water availability (drainage) in basins within the Mississippi. Areas with large vadose zones can have changes in the vadose zones dominate the changes in TWSA.

*Response: We will add a statement to clarify that most of the variability in TWS in this region comes from soil water storage changes, including the vadose zone. However, we did not understand the reviewer comment about "water availability" not being related to TWSA.*

***Text will be modified at P4 L1-2:*** *Following work by Kim et al. (2009), we focus on the fact that most summer storage variability in the Mississippi River basin is not due to surface water storage, but instead to sub-surface storage (including vadose zone).*

**Comment 7)P7 L6-11:** This is a summary not conclusion. The conclusions need to be

bolstered, at the moment they are quite weak.

*Response: We will change the text to clarify that this statement is a summary, not a conclusion. To that end, we will add the phrase "in summary". Also, we now briefly summarize the results and the motivation for the study in the conclusions.*

***Text will be modified at P7 L6-20:*** *Given the importance of knowing how much water is available for societal demands and the complexity to measure this quantity with traditional methods, the primary goals of this research are to estimate total drainable water storage and the fraction of baseflow in the Mississippi River basin using remotely sensed measurements.*

*In summary, our approach focuses on non-winter months (Apr-Nov) for the period of April 2002 through October 2014 for 12 watersheds distributed throughout the Mississippi Basin. A forward-looking, low flow filter is used to approximate baseflow from measured discharges. Exponential relationships between discharge and NASA's GRACE total water storage anomalies are developed for all 12 sub-areas. The relationships show that the fraction of baseflow in the sub-basins varies from 52 to 75% regionally. The provided approach can be used to provide estimates of drainable water storage for watersheds larger than roughly 200,000 km2 using only measurements derived from the GRACE mission and monthly streamflow gage measurements. For the Mississippi River Basin in the period of 2002 to 2014, the drainable water storage ranged from 2,900 ± 400 km3 to 3,600 ± 400 km3.*

*Since we base our analysis on observed quantities, a certain level of empiricism is required to validate the methodology. Still, we believe that this analysis is an initial step towards further understanding the relationship between storage and discharge. Future research is recommended to: investigate the effects of temporal subsampling in developing Q-S relationships; explore additional methods for estimating baseflow values for each increasing storage change value; explore additional methods to estimate So with and/or without measured discharges; and integrate winter months into*

[Figure]

the analysis to characterize year-round discharge-storage relationships. Our long-term goal is to estimate discharge (e.g., baseflow) without gauge measurements to characterize and model hydrologic and ecological cycles in regions with limited or no in-situ measurements.

**Comment 8)P7 L13:** You didn't just use TWSA, you used Q as well.

*Response: That's true – the method offers an approach based on coupled TWSA and Q measurements.*

**Text will be modified at P7 L13:** *The provided approach can be used to provide estimates of drainable water storage for watersheds larger than roughly 200,000 km2 using measurements derived from the GRACE mission and monthly streamflow gage measurements.*

**Comment 9)Figure 3:** Are these regressions significant? Include axis labels.

*Response: All relationships are significant at a 99% confidence interval (p-value < 0.00001), based on t-test. The axis labels are described in the figure caption to avoid redundancy in the figure.*

**Text will be modified at the caption at Figure 3:** *Figure 3: Non-winter (Apr-Oct) monthly observed discharge (Qo; y-axis in units of cm) and storage (S, x-axis in units of cm represented by TWSAs); the lines represent the relationship between observed discharge (blue) or baseflow (red) and storage. The plots IDs correspond to the site IDs listed Table 1 and shown in Figure 1. All relationships are significant at a 99% confidence interval (p-value < 0.00001), based on t-test.*

**Comment 10)Figure 4 (and within text):** This insinuates that drainable storage didn't change with time. How do you justify this in such a dynamic basin?

*Response: Drainable storage relates to the long-term mean storage capacity of the basin. It is time-invariant by definition. While this may evolve in the long-term due to geological or land use changes, we offer a first estimate over the years 2002-2014.*

**References**

*Brookfield, A. E., Hill, M. C., Rodell, M., Loomis, B. D., Stotler, R. L., Porter, M. E., and Bohling, G. C.: In Situ and GRACE-Based Groundwater Observations: Similarities, Discrepancies, and Evaluation in the High Plains Aquifer in Kansas, Water Resources Research, 54, 8034-8044, doi: 10.1029/2018wr023836, 2018.*

*David, C. H., Famiglietti, J. S., Yang, Z.-L., and Eijkhout, V.: Enhanced fixed-size parallel speedup with the Muskingum method using a trans-boundary approach and a large subbasins approximation, Water Resources Research, 51, 7547-7571, doi: 10.1002/2014wr016650, 2015. Kim, H., Yeh, P. J. F., Oki, T., and Kanae, S.: Role of rivers in the seasonal variations of terrestrial water storage over global basins, Geophysical Research Letters, 36, doi: 10.1029/2009GL039006, 2009.*

*Nie, N., Zhang, W., Chen, H., and Guo, H.: A Global Hydrological Drought Index Dataset Based on Gravity Recovery and Climate Experiment (GRACE) Data, Water Resources Management, 32, 1275-1290, doi: 10.1007/s11269-017-1869-1, 2018.*

*Rodell, M., Chao, B. F., Au, A. Y., Kimball, J. S., and McDonald, K. C.: Global Biomass Variation and Its Geodynamic Effects: 1982–98, Earth Interactions, 9, 1-19, doi: 10.1175/ei126.1, 2005.*

*Rodell, M., Chen, J., Kato, H., Famiglietti, J. S., Nigro, J., and Wilson, C. R.: Estimating groundwater storage changes in the Mississippi River basin (USA) using GRACE, Hydrogeology Journal, 15, 159-166, doi: 10.1007/s10040-006-0103-7, 2007.*

*Scanlon, B. R., Faunt, C. C., Longuevergne, L., Reedy, R. C., Alley, W. M., McGuire, V. L., and McMahon, P. B.: Groundwater depletion and sustainability of irrigation in the US High Plains and Central Valley, Proceedings of the National Academy of Sciences, 109, 9320-9325, doi: 10.1073/pnas.1200311109, 2012.*

---

## Author Comment (AC2) · 10 May 2019

**Author's comments – Referee 2**

The paper aims to estimate the amount of drainable water storage in a basin using GRACE satellite and streamflow data. They develop a forward-looking, low flow filter to isolate base flow; while transforming GRACE based storage anomalies to provide estimates of absolute drainable water storage in the Mississippi River Basin. The work is of interest and suitable for this journal as it deals with a fundamental aspect of hydrology, and provides useful technique to investigate storage-outflow relationships of large watersheds. Overall, the paper is written well and the figures are clear. The pa-

per, however, would benefit from some major revisions, especially with regards to the introduction and methods section. For this reason, I suggest the editor consider the revisions suggested below prior to making a decision on this manuscript.

*Response: We thank the reviewer for the valuable comments and attention to detail. Responses to your concerns are provided below, along with suggestions for changes in the manuscript for a posterior resubmission.*

**Major comments:**

**Comment 1** - The authors reference other studies that have used remote sensing to estimate water storage in basin; after looking at the titles of those journal articles, it seems that at least 2 of those studies (Tourian et.al., 2018; Riegger, 2018) have attempted to estimate total drainable water storage in a basin using GRACE data. How are the methods used in the present study different from those analysis? If the methods are different, then why was a different method developed? If there is a significant overlap in methods, then what is the novel contribution of this study? The answers to these questions should be clearly integrated into the introduction, as the original contributions of the authors seem unclear.

*Response: Regarding the reviewer's suggestion concerning two previous studies, we have made some important changes to the manuscript to highlight the differences. Note that the Riegger (2018) article has not been peer-reviewed, as it was only accepted as a discussion paper. On the premise that such a paper may not pass peer-review, we avoid specific discussion of that paper and its methods here. Tourian et al. (2018) was the first study to estimate a total drainable water storage from a large river basin. This was done by estimating a linear relationship between the storage variability with the discharge at the mouth and applying a phase shift between the two timeseries using a Hilbert transform. In the current work, we have used a different approach, which allows for non-linearity in the storage-discharge relationship by treating only the case of storage driven flow (or baseflow). This is done by applying a traditional hy-*

drological analysis technique called baseflow recession. In contrast to Tourian et al. (2018) this is an augmentation and refinement of the previous technique, applied over a new and different study domain.

***Text will be modified at P4 L10-15:*** *We define drainable water storage as "the volume of water in a basin that is connected to streamflow and would drain out of the basin as time went towards infinity with no additional precipitation inputs". Tourian et al. (2018) was the first study to estimate a total drainable water storage from a large river basin. This was done by estimating a linear relationship between the storage variability with the discharge at the mouth and applying a phase shift between the two time-series using a Hilbert transform. Here, to characterize the drainable storage from the sub-basins, GRACE TWSAs are transformed into drainable water storages (i.e., not anomalies) using the derived discharge-TWSA relationships. Applying baseflow recession allows for non-linearity in the discharge-storage relationship by treating only the case of storage driven flow (baseflow). For the first time, we demonstrate the direct relationship between storage and discharge on a basin and sub-basin scale, we estimate parameters in the baseflow recession equation and we give the first estimate of total drainable water storage that has never been estimated using only observations.*

**Comment 2** - As pointed out by referee1, the methods section needs to be written better especially with regards to how Qb was estimated. It seems unclear as to which "20% of the number of pairs (months)" were used to get the minimum value. Also, it would be useful to include a figure that shows the sensitivity of the model to n in the supplementary document to solidify that 20% was indeed a correct forward looking limit.

***Response:*** *Since both reviewers pointed out that our methods to estimate Qb are not entirely clear, we have rewritten that section. We can add a figure with the n sensitivity analysis if necessary.*

***Text will be modified at P4 L10-15:*** *To build the Qb-S relationship, the Qo-S fixed*

*paired series is sorted from the minimum to maximum value of S. Because Qo is assumed to increase with S, Qb for a given S is set to the forward-looking minimum Qo. Next, a Qb value is estimated for each S, based on minimum measured values of Qo: (equation 1) where n is the number of forward-looking values remaining in the paired series. In other words, the filter looks at the next n Qo values paired to the next n larger S values, selecting the minimum Qo as baseflow. The n value can be subjective depending on the series size. Here, we used 20% of the number of pairs (18 months), after analyzing the model's sensitivity to n.*

**Comment 3** - The justification of using Q-S relationship in a highly regulated systems (like the Missouri River) needs to be added. Can the storage values obtained in these systems still be considered as the total drainable water storage? How do the reservoir operational policies affect the low flow values obtained? It might be useful to go deeper into one of these regulated systems to explain why the estimates obtained are still useful/valid there.

*Response: The reviewer's comment is important and deserves some explanation. To quote our answer to Reviewer 1: "For the larger river basins and their major rivers, streamflow shows a first-order response to precipitation and storage changes within the basin, which justifies the first-order validity of our methodology. The higher order "errors" introduced in our approach due to the misrepresentation of natural discharge would affect our recession approach, but there are challenges in quantifying these errors directly. Considering the timescales of a rainstorm and runoff event, we assume that most reservoir operations would only significantly affect the downstream (i.e. large river) discharge due to small reservoir operations within a finite time-span and with finite storage volume (i.e. approximately 5-10% of the discharge signal). Studies on numerical modeling of the Mississippi river and the estimated effect of diversions and reservoirs at the gage support these estimates (e.g. David et al., 2015)."*

*Therefore, in general, the drainage water storage in regulated systems is likely 5-10% larger than reflected herein. However, this effect is magnified for the Missouri River*

*basin due to the heavy regulation, increasing the uncertainty of the simulated value.*

***Text will be modified at P5-6 L31-6:*** *Based on a qualitative assessment, $\beta$ appears to decrease as the amount of water regulation increases. For example, the Missouri River is known to be highly regulated and the associated $\beta$ values are noticeably lower than those for the upper Mississippi and Ohio Rivers. In a regulated system, basin storage can increase with little change in river discharge because water is being stored in lakes/reservoirs. In this case, the Missouri river has several very large reservoirs (e.g., Lake Oahe, Lake Sakakawea, Fort Peck Lake), which may explain the relative lower relation between Q-S (panels 7-9 at Fig. 3). This is one of this method's limitations, creating an uncertainty from the inability to include specific basin characteristics. For this reason, the relationships for heavy regulated rivers only reflects reservoir storage availability observed during the study period and that drainage water storage in these systems is likely larger than reflected herein. Of interest is the difference in $\beta o$ and $\beta b$ along the Missouri River, where $\beta b$ is roughly 35-62% of $\beta o$ as compared to the other rivers where $\beta b$ is 84-110% of $\beta o$. This difference, which is due to disproportionally lower $\beta o$ values for the Missouri River, suggests that in regulated systems storage changes are mitigated more for baseflow as compared to event-flow conditions (Fig. 3).*

**Comment 4** - The authors claim that the total drainable storage volumes they obtain cannot be validated. Can large-scale hydrological models like PCR-GLOBWB be used to obtain similar values? There should be some acknowledgement of the ability or inability of large-scale hydrological models to estimate a similar value.

***Response:*** *We initially thought of this as well. However, many large-scale models (e.g., those included in NASA's GLDAS system; PCR-GLOBWB) are not fully coupled with groundwater models nor do they include spatially varying soil depth. Thus, the comparison would not be a direct comparison. Previous studies by Houburg et al. (2012) and Scanlon et al (2018), highlight the impacts of model structural errors on the ability to represent the GRACE-observed storage variability.*
*Text will be modified at P7 L2: These values cannot be validated since there are no current measurements of such an amount. Most large-scale models (e.g. PCR-GLOBWB, van Beek and Bierkens, 2009) are not fully coupled with groundwater models and contain structural errors on the ability to represent the GRACE-observed storage variability (Houborg et al., 2012;Scanlon et al., 2018). Thus, the comparison would not be direct.*

**Comment 5** - The conclusions section currently seems to be a summary of the methods used in the study and the scope of future work. This section should be expanded further to include some of the results obtained, as well as a discussion of why/where it is important to know the total drainable storage of a basin.

*Response: This point is well taken. We note that the motivation for the study was provided in the introduction. The discussion of the results and the results are provided in those respective sections and in the abstract. However, following the reviewer's suggestion, we now briefly summarize the results and the motivation for the study in the conclusions.*

*Text will be modified at P7 L6-20: Given the importance of knowing how much water is available for societal demands and the complexity to measure this quantity with traditional methods, the primary goals of this research are to estimate total drainable water storage and the fraction of baseflow in the Mississippi River basin using remotely sensed measurements.*

*In summary, our approach focuses on non-winter months (Apr-Nov) for the period of April 2002 through October 2014 for 12 watersheds distributed throughout the Mississippi Basin. A forward-looking, low flow filter is used to approximate baseflow from measured discharges. Exponential relationships between discharge and NASA's GRACE total water storage anomalies are developed for all 12 sub-areas. The relationships show that the fraction of baseflow in the sub-basins varies from 52 to 75% regionally. The provided approach can be used to provide estimates of drainable wa-*

*ter storage for watersheds larger than roughly 200,000 km2 using only measurements derived from the GRACE mission and monthly streamflow gage measurements. For the Mississippi River Basin in the period of 2002 to 2014, the drainable water storage ranged from 2,900 ± 400 km3 to 3,600 ± 400 km3.*

*Since we base our analysis on observed quantities, a certain level of empiricism is required to validate the methodology. Still, we believe that this analysis is an initial step towards further understanding the relationship between storage and discharge. Future research is recommended to: investigate the effects of temporal subsampling in developing Q-S relationships; explore additional methods for estimating baseflow values for each increasing storage change value; explore additional methods to estimate So with and/or without measured discharges; and integrate winter months into the analysis to characterize year-round discharge-storage relationships. Our long-term goal is to estimate discharge (e.g., baseflow) without gauge measurements to characterize and model hydrologic and ecological cycles in regions with limited or no in-situ measurements.*

**Minor comments:**

**Comment P1 L24-26:** The sentence does not read correctly. I suggest having a separate sentence to describe/summarize the remote sensing that has contributed to estimating watershed storage.

*Response: We suggest the change:*

**Text will be modified at P1 L24-26:** *Despite the importance of characterizing watershed storage, relatively little work has been done to understand the relationship between storage and discharge. Most of the existing work is based on remotely-sensed observations of storage (eg., Riegger and Tourian, 2014;Reager et al., 2014;Sproles et al., 2015;Tourian et al., 2018;Riegger, 2018).*

**Comment P2 L11:** "the desire" seems redundant. Suggestion: "The motivation was to

create a functional relationship. . ..."

*Response: We agree with the reviewer. This sentence will be fixed in the next manuscript version.*

**Text will be modified at P2 L11-12:** *The motivation was to create a functional relationship between discharge and storage that could then be used to model discharge using only precipitation and evapotranspiration data.*

**Comment Figure 1:** It would be useful to include the sub-basin boundaries on the map to help orient the readers

*Response: This is an excellent suggestion. We now shade the Ohio, Upper Miss and Missouri basins in our revised Figure 1.*

**Comment P4 L25-34:** While it is implied that the authors use this expression to estimate the absolute water storage, it might be useful to explicitly state that here.

*Response: We suggest adding the word water in the specified section.*

**Text will be modified at P4 L25-34:** *To transform TWSA into an absolute water storage value, referenced herein as drainable storage (Se) that directly influences discharge, a storage offset must best estimated. For example, Riegger and Tourian (2014) proposed a definition of time-dependent absolute water storage Se(t), using Eq. (3): (equation 3) where ðĺŚĞðĺŚŁðĺŚĘðĺŘť(ðĺŚą) is the monthly storage anomaly and So is an unknown constant storage offset. So only shifts the Se(t) series without impacting its temporal variability. This storage offset cannot be measured directly but should correspond to the long-term mean water storage for the region of interest. Based on the assumption that baseflow is driven by storage (Se) and therefore a linear function of storage, the relationship between discharge and TWSA can provide insights for estimating the representative So value, which provides an opportunity to estimate drainable storage.*

**Comment P5 L3-7:** It would be more useful to integrate this paragraph into the meth-
ods section as there seems to be no results here.

*Response: We agree with the reviewer. This paragraph will be moved to the end of the Methods section.*

**Comment P5 L24:** Replace with "which corresponds to the mean"

*Response: We agree with the reviewer. The changes will be incorporated to the new manuscript version.*

**Text will be modified at P5 L24-25:** *which corresponds to the mean storage observed during the GRACE period.*

**Comment P7 L2:** Replace with "of such an amount"

*Response: We agree with the reviewer. The changes will be incorporated to the new manuscript version.*

**Text will be modified at P7 L2:** *These values cannot be validated since there are no current measurements of such an amount.*

**References:**

*David, C. H., Famiglietti, J. S., Yang, Z.-L., and Eijkhout, V.: Enhanced fixed-size parallel speedup with the Muskingum method using a trans-boundary approach and a large subbasins approximation, Water Resources Research, 51, 7547-7571, doi: 10.1002/2014wr016650, 2015.*

*Houborg, R., Rodell, M., Li, B., Reichle, R., and Zaitchik, B. F.: Drought indicators based on model-assimilated Gravity Recovery and Climate Experiment (GRACE) terrestrial water storage observations, Water Resources Research, 48, doi: 10.1029/2011wr011291, 2012.*

*Riegger, J.: Quantification of Drainable Water Storage Volumes in Catchments and in River Networks on Global Scales using the GRACE and/or River Runoff, Hydrol. Earth*

*Syst. Sci. Discuss., 2018, 1-27, doi: 10.5194/hess-2018-38, 2018.*

*Scanlon, B. R., Zhang, Z., Save, H., Sun, A. Y., Müller Schmied, H., van Beek, L. P. H., Wiese, D. N., Wada, Y., Long, D., Reedy, R. C., Longuevergne, L., Döll, P., and Bierkens, M. F. P.: Global models underestimate large decadal declining and rising water storage trends relative to GRACE satellite data, Proceedings of the National Academy of Sciences, 115, E1080-E1089, doi: 10.1073/pnas.1704665115, 2018.*

*Tourian, M. J., Reager, J. T., and Sneeuw, N.: The Total Drainable Water Storage of the Amazon River Basin: A First Estimate Using GRACE, Water Resources Research, 54, 3290-3312, doi: 10.1029/2017WR021674, 2018.*

*van Beek, L., and Bierkens, M. F.: The global hydrological model PCR-GLOBWB: conceptualization, parameterization and verification, Department of Physical Geography, Utrecht University, Utrecht, The Netherlands, 2009.*

---

## Author Response (AR1)

**Author's Response – Referee 1**

General Comments: This manuscript describes a method of relating streamflow measurements and terrestrial water storage anomalies (TWSA) from GRACE data products to estimate the drainable storage of several Mississippi River sub basins. This research is current, relevant, and of interest to the readers of HESS. The manuscript was well written and organized, and I enjoyed reading it. I have a couple of concerns with the fundamental concepts that underpin this research that require further explanation from the authors, as described under 'specific comments' below. In addition, I have further minor/editorial comments provided under 'technical corrections' below. Overall, I think this manuscript should be returned to the authors for major revisions.

*Response: We thank the reviewer for the support and the feedback to improve our manuscript. Detailed responses to your concerns are provided below, along with the changes in the manuscript for resubmission.*

**Specific Comments:**

**Comment 1:** First of all, the methodology estimating Qb is not clear. The authors state that the Qo-S pairs are ordered by size of S, then Qb is the 'forward-looking minimum' of Qo. Is this forward in time, or just in this ordered pairing from low to high S? I assume forward in time, because you can't simply ignore the order of events (a low S cannot be the result of a low Qo that won't occur for several months). In addition, the text says that Qb is estimated as a fraction of Qo using equation 1, yet equation 1 contains no metric for this fraction. Either this is the incorrect equation, or the term 'fraction' is used in error.

*Response: We understand your concern, and the lack of clarity on the estimation method for Qb. To reiterate, the equation we apply assumes that 'baseflow' comprises the storage-driven portion of the streamflow, but that there are other portions of streamflow contributed by surface runoff generation (i.e. not 'baseflow'). The discharge at any time is some combination of those two processes, and while the baseflow varies depending only on storage, the surface runoff varies based on other processes including precipitation rate, land cover type, and surface soil saturation, and tends to vary more rapidly. After pairing GRACE TWSA with observed monthly discharge for each basin, the paired time series is sorted from minimum to maximum value of S. We are not ignoring the order of events, because even after sorting, the pairs still correspond to the same month. Then, for each pair, the filter looks at the next 18 discharge values (next 18 Qo paired to the next 18 larger S values), selecting the minimum value as the baseflow. Because the storage driven baseflow represents a partial component of the total streamflow corresponding to the minimum surface runoff (or ideally zero surface runoff) situation, we aim to find the case when the non-baseflow signal is minimized and baseflow dominates. The TWSA-discharge pair doesn't change, but this method selects a minimum discharge value that was measured (realistic) for a similar magnitude (in the same magnitude bin) of storage, in hopes that this will best represent only the baseflow portion of the discharge signal.*

*You are correct about the use of the word "fraction".*

*Text was modified at P4 L23-30.*

**Comment 2:** The second concern is related to the temporal resolution of the data with respect to equation 1. GRACE data represent the TWSA on the particular day(s) the measurements were taken, and not the monthly high/low/average. Qo is defined as the mean monthly observed discharge. Thus, equation 1 is dependent upon pairing of an instantaneous value with a mean value. While some work indicates that TWSA variability is largely not due to surface water storage (storage that can fluctuate greatly with time), some evaluation of the variability of Q throughout each month should be considered before applying equation 1.

*Response: The reviewer is correct in that we assume that an average discharge value for the whole month is analogous to, or corresponds to, the GRACE monthly solution. In fact, a monthly GRACE solution may integrate temporal information from several ground tracks through the study region into the monthly gravity field, and each of those ground tracks could have been recorded on a separate day of that month. So, it is also erroneous to say that a monthly GRACE solution represents a single day over the study region. If the study region is relatively large, like the river basins here, it is highly likely that several samples of information throughout the month are included in the solution.*

*With this in mind, the issue the reviewer has identified is worth mentioning, but there is no clear path to overcome this issue at this time. It is not clear how the daily analysis of discharge would offer any insights, in terms of the fraction of discharge that is driver by baseflow or surface water, and an (e.g.) statistical analysis of the discharge time series alone with no complimentary information from GRACE would not really offer a new methodological approach.*

*As the reviewer mentioned, following work by Kim et al. (2009) to partition variability in the GRACE signal in global river basins, we focus on the fact that most summer storage variability in the Mississippi River basin is primarily not due to surface water storage, but instead to sub-surface storage in soils, and therefore lend themselves well to a baseflow recession analysis.*

*As such, we now mention this point as a caveat of the study, and text to this effect appears in the methods section.*

*Text was modified at P4 L15-19.*

**Comment 3:** Thirdly, while only considering nonwinter storage variability simplifies the analysis with respect to snow accumulation and events, it does complicate the issue with respect to vegetation growth. The Mississippi basin is a large agricultural area, and a change in mass due to the increase in vegetation over the growing season should be addressed in this work. Along similar lines, I would be interested to know how much groundwater pumping takes place within each sub basin, and if that contributes significantly to changes in TWSA.

*Response: Our study was focused on non-winter storage variability by necessity: this approach provides the best look at the storage-discharge relationship without the complication of freeze-thaw processes on both storage and discharge. Based on global maps of vegetation biomass (Rodell et al., 2005), Rodell et al. (2007) affirms that the seasonal and interannual biomass variations are typically smaller than the*

*uncertainty in the GRACE TWSA measurements. While still a source of uncertainty that should be cited in our work (and will be fixed in the final manuscript), this also holds true for the Mississippi River basin.*

*Groundwater pumping is really a separate topic. Significant pumping does occur in the High Plains aquifer, which is a shallow-water-table aquifer. As such, in the case for which changes in storage from groundwater pumping would lower the water table, those storage changes would still be linked directly to baseflow generation. So those storage changes would be generally consistent with the current approach and hypothesis. In other words, the portions of the basin which are experiencing water table decline due to human activities would still exhibit the same general storage-discharge relationship, and while an in-depth analysis of groundwater pumping activities would theoretically interesting, it would not augment the results of our study, nor provide coherent insight on our results. There are already several studies on the High Plains aquifer using GRACE to monitor groundwater changes (e.g. Scanlon et al., 2012;Brookfield et al., 2018;Nie et al., 2018).*

***Text was added at P3-4 L30-5.***

**Comment 4:** Finally, while the authors address the issue of reservoir storage and releases and their influence over Q I think further work is needed to discuss how the Q-S relationships can still hold in these environments. If the flow of the stream is dependent upon reservoir releases they would not necessarily reflect the basin's storage (e.g. we can have a large reservoir release when groundwater levels (a reflection of baseflow) and drainable storage, are low), so how can the Q-S relationship still hold? Many reservoirs in the Mississippi basin are driven by downstream user demands and are not a reflection of what the natural flow conditions would be.

***Response:*** *To say that the streamflow is dependent upon reservoir releases is true to some extent for the smaller tributaries in the study domain and we should do a better job of clarifying our assumptions. For the larger river basins and their major rivers, streamflow shows a first-order response to precipitation and storage changes within the basin. The higher order "errors" introduced in our approach due to the misrepresentation of natural discharge would affect our recession approach, but there are challenges in quantifying these errors directly. Considering the timescales of a rain storm and runoff event, we assume that most reservoir operations would only significantly affect the downstream (i.e. large river) discharge due to small reservoir operations within a finite time-span and with finite storage volume (i.e. approximately 5-10% of the discharge signal). Studies on numerical modeling of the Mississippi river and the estimated effect of diversions and reservoirs at the gage support these estimates (e.g. David et al., 2015).*

*However, the same study by David et al. (2015) points that heavy regulation causes longer residence times and dampening of the flow, and the lack of representation of storage processes in reservoirs caused poor simulation of modeled flow in the Missouri River basin. This is one of this method's limitations, creating an uncertainty from the inability to include specific basin characteristics. This effect is a lot more pronounced for the Missouri River sub-basins than it is for others reflecting on a relative lower relation between Q-S. This effect is evidenced in the $R^2$ in the panels 7-9 at Figure 3.*

***Text was modified at P6 L20-23.***

**Technical Comments**

**Comment 1)P3 L2:** You provide an estimate of drainable basin storage, not total basin storage.

**Comment 2)P3 L16:** 'smaller size' not 'inferior size'

**Comment 3)P4 L5:** Perhaps indicate that you focus on storage anomalies because it is not possible to quantify absolute storage with GRACE data.

*Response: Comments 1, 2, 3 are very pertinent, they were easily incorporate into our text.*

**Text was modified at P3 L4-5.**

**Text was modified at P3 L19.**

**Text was modified at P4 L19.**

**Comment 4)P3-4, L31-1:** This sentence is redundant

**Comment 5)P5 L4-5:** This should not be a surprise since you derived S from TWSA.

*Comments 4, 5: We understand how these sentences can be seen as redundant, we initially included them for enhanced clarity and rewrote them in our resubmission.*

**Text was modified at P4 L8-10.**

**Text was modified at P5 L12-15.**

**Comment 6)P4 L2**: Recent research supports the conclusion that TWSAs are not due to surface anomalies, but also indicates that TWSAs are not related to water availability (drainage) in basins within the Mississippi. Areas with large vadose zones can have changes in the vadose zones dominate the changes in TWSA.

*Response: We added a statement to clarify that most of the variability in TWS in this region comes from soil water storage changes, including the vadose zone. However, we did not understand the reviewer comment about "water availability" not being related to TWSA.*

**Text was modified at P4 L12.**

**Comment 7)P7 L6-11:** This is a summary not conclusion. The conclusions need to be bolstered, at the moment they are quite weak.

*Response: We changed the text to clarify that this statement is a summary, not a conclusion. To that end, we added the phrase "in summary". Also, we now briefly summarize the results and the motivation for the study in the conclusions.*

**Text was modified at P7-8 L28-6.**

**Comment 8)P7 L13:** You didn't just use TWSA, you used Q as well.

*Response: That's true – the method offers an approach based on coupled TWSA and Q measurements.*

*Text was modified at P8 L4.*

**Comment 9)Figure 3:** Are these regressions significant? Include axis labels.

*Response: All relationships are significant at a 99% confidence interval (p-value < 0.00001), based on t-test. The axis labels are described in the figure caption to avoid redundancy in the figure.*

***Text was modified at the caption at Figure 3.***

**Comment 10)Figure 4** (and within text): This insinuates that drainable storage didn't change with time. How do you justify this in such a dynamic basin?

*Response: Drainable storage relates to the long-term mean storage capacity of the basin. It is time-invariant by definition. While this may evolve in the long-term due to geological or land use changes, we offer a first estimate over the years 2002-2014.*

**Author's comments – Referee 2**

The paper aims to estimate the amount of drainable water storage in a basin using GRACE satellite and streamflow data. They develop a forward-looking, low flow filter to isolate base flow; while transforming GRACE based storage anomalies to provide estimates of absolute drainable water storage in the Mississippi River Basin. The work is of interest and suitable for this journal as it deals with a fundamental aspect of hydrology, and provides useful technique to investigate storage-outflow relationships of large watersheds. Overall, the paper is written well and the figures are clear. The paper, however, would benefit from some major revisions, especially with regards to the introduction and methods section. For this reason, I suggest the editor consider the revisions suggested below prior to making a decision on this manuscript.

*Response: We thank the reviewer for the valuable comments and attention to detail. Responses to your concerns are provided below, along with the changes in the manuscript for the resubmission.*

**Major comments:**

**Comment 1 -** The authors reference other studies that have used remote sensing to estimate water storage in basin; after looking at the titles of those journal articles, it seems that at least 2 of those studies Tourian, 2018; Riegger, 2018) have attempted to estimate total drainable water storage in a basin using GRACE data. How are the methods used in the present study different from those analysis? If the methods are different, then why was a different method developed? If there is a significant overlap in methods, then what is the novel contribution of this study? The answers to these questions should be clearly integrated into the introduction, as the original contributions of the authors seem unclear.

*Response: Regarding the reviewer's suggestion concerning two previous studies, we have made some important changes to the manuscript to highlight the differences. Note that the Riegger (2018) article has not been peer-reviewed, as it was only accepted as a discussion paper. On the premise that such a paper*

*may not pass peer-review, we avoid specific discussion of that paper and its methods here. Tourian et al. (2018) was the first study to estimate a total drainable water storage from a large river basin. This was done by estimating a linear relationship between the storage variability with the discharge at the mouth and applying a phase shift between the two timeseries using a Hilbert transform. In the current work, we have used a different approach, which allows for non-linearity in the storage-discharge relationship by treating only the case of storage driven flow (or baseflow). This is done by applying a traditional hydrological analysis technique called baseflow recession. In contrast to Tourian et al. (2018) this is an augmentation and refinement of the previous technique, applied over a new and different study domain.*

**Text was modified at P2-3 L30-5.**

**Comment 2 -** As pointed out by referee#1, the methods section needs to be written better especially with regards to how Qb was estimated. It seems unclear as to which "20% of the number of pairs (months)" were used to get the minimum value. Also, it would be useful to include a figure that shows the sensitivity of the model to n in the supplementary document to solidify that 20% was indeed a correct forward looking limit.

*Response: Since both reviewers pointed out that our methods to estimate Qb are not entirely clear, we have rewritten that section. We added a figure with the n sensitivity analysis in the Supporting Information document.*

**Text was modified at P4 L25-30.**

**Comment 3 -** The justification of using Q-S relationship in a highly regulated systems (like the Missouri River) needs to be added. Can the storage values obtained in these systems still be considered as the total drainable water storage? How do the reservoir operational policies affect the low flow values obtained? It might be useful to go deeper into one of these regulated systems to explain why the estimates obtained are still useful/valid there.

*Response: The reviewer's comment is important and deserves some explanation. To quote our answer to Reviewer #1: "For the larger river basins and their major rivers, streamflow shows a first-order response to precipitation and storage changes within the basin, which justifies the first-order validity of our methodology. The higher order "errors" introduced in our approach due to the misrepresentation of natural discharge would affect our recession approach, but there are challenges in quantifying these errors directly. Considering the timescales of a rainstorm and runoff event, we assume that most reservoir operations would only significantly affect the downstream (i.e. large river) discharge due to small reservoir operations within a finite time-span and with finite storage volume (i.e. approximately 5-10% of the discharge signal). Studies on numerical modeling of the Mississippi river and the estimated effect of diversions and reservoirs at the gage support these estimates (e.g. David et al., 2015)."*

*Therefore, in general, the drainage water storage in regulated systems is likely 5-10% larger than reflected herein. However, this effect is magnified for the Missouri River basin due to the heavy regulation, increasing the uncertainty of the simulated value.*

**Text was modified at P6 L20-23.**

**Comment 4 -** The authors claim that the total drainable storage volumes they obtain cannot be validated. Can large-scale hydrological models like PCR-GLOBWB be used to obtain similar values? There should be some acknowledgement of the ability or inability of large-scale hydrological models to estimate a similar value.

*Response: We initially thought of this as well. However, many large-scale models (e.g., those included in NASA's GLDAS system; PCR-GLOBWB) are not fully coupled with groundwater models nor do they include spatially varying soil depth. Thus, the comparison would not be a direct comparison. Previous studies by Houborg et al. (2012) and Scanlon et al. (2018), highlight the impacts of model structural errors on the ability to represent the GRACE-observed storage variability.*

**Text was modified at P7 L22-25.**

**Comment 5 -** The conclusions section currently seems to be a summary of the methods used in the study and the scope of future work. This section should be expanded further to include some of the results obtained, as well as a discussion of why/where it is important to know the total drainable storage of a basin.

*Response: This point is well taken. We note that the motivation for the study was provided in the introduction. The discussion of the results and the results are provided in those respective sections and in the abstract.*

*However, following the reviewer's suggestion, we now briefly summarize the results and the motivation for the study in the conclusions.*

**Text was modified at P7-8 L28-6.**

**Minor comments:**

**Comment P1 L24-26:** The sentence does not read correctly. I suggest having a separate sentence to describe/summarize the remote sensing that has contributed to estimating watershed storage.

*Response: We modified the sentence for clarity.*

**Text was modified at P1 L25.**

**Comment P2 L11:** "the desire" seems redundant. Suggestion: "The motivation was to create a functional relationship. . ..."

*Response: We agree with the reviewer.*

**Text was modified at P2 L11.**

**Comment Figure 1:** It would be useful to include the sub-basin boundaries on the map to help orient the readers

*Response: This is an excellent suggestion.*

**We now shade the Ohio, Upper Mississippi and Missouri basins in our revised Figure 1. (P13)**

**Comment P4 L25-34:** While it is implied that the authors use this expression to estimate the absolute water storage, it might be useful to explicitly state that here.

*Response: We suggest adding the word water in the specified section.*

*Text was modified at P5 L7-16.*

**Comment P5 L3-7:** It would be more useful to integrate this paragraph into the methods section as there seems to be no results here.

*Response: We agree with the reviewer. This paragraph was moved to the end of the Methods section.*

*Text was moved to P5 L12-15.*

**Comment P5 L24:** Replace with "which corresponds to the mean"

*Response: We agree with the reviewer. The changes are now incorporated to the new manuscript version.*

*Text was modified at P6 L9.*

**Comment P7 L2:** Replace with "of such an amount"

*Response: We agree with the reviewer. The changes are now incorporated to the new manuscript version.*

*Text was modified at P7 L22.*

[revised manuscript text omitted]
. Figure S1 shows the sensitivity analysis performed to select the number of forward-looking values (n) to be used in the low-flow filter method. Table S1 includes the $\alpha$ and $\beta$ parameters for all 12 stations, which can be seen at Figure 3. Table S2 lists the storage offsets used to convert GRACE TWSA into absolute drainable storage values, used to create Figure 4.

[Figure]

**Figure S1.** Sensitivity analysis of the number of forward-looking values (*n*) used in the low-flow filter method. The baseflow fraction of storage offset (So) and β coefficient were estimated for 6 different *n* filter scenarios (5, 10, 15, 18, 20, 25) for each station. For clarity, stations with similar relationships were aggregated in two groups (1-6,12 and 7-11). In general, the baseflow fraction of both parameters stabilizes for an *n* value of roughly 18.

**Table S1.** Resulting storage-discharge coefficients for overall discharge and baseflow with associated $R^2$ values.

| ID | $\alpha_o$ | $\beta_o$ | $R^2$ | $\alpha_b$ | $\beta_b$ | $R^2$ |
|----|-----------|-----------|-------|-----------|-----------|-------|
| 1 | 2.7230 | 0.1048 | 0.78 | 1.4223 | 0.0876 | 0.85 |
| 2 | 2.4810 | 0.1108 | 0.80 | 1.3029 | 0.1022 | 0.88 |
| 3 | 2.7214 | 0.0759 | 0.75 | 1.5374 | 0.0761 | 0.92 |
| 4 | 1.8031 | 0.0875 | 0.59 | 1.1570 | 0.0781 | 0.88 |
| 5 | 1.8557 | 0.0999 | 0.66 | 1.2188 | 0.1066 | 0.91 |
| 6 | 1.9825 | 0.0992 | 0.74 | 1.2844 | 0.1055 | 0.91 |
| 7 | 0.2209 | 0.0511 | 0.46 | 0.1534 | 0.0319 | 0.72 |
| 8 | 0.2616 | 0.0466 | 0.40 | 0.1834 | 0.0162 | 0.46 |
| 9 | 0.2848 | 0.0441 | 0.45 | 0.2133 | 0.0184 | 0.71 |
| 10 | 0.2854 | 0.0508 | 0.51 | 0.2142 | 0.0248 | 0.91 |
| 11 | 0.4224 | 0.0691 | 0.54 | 0.2659 | 0.0358 | 0.90 |
| 12 | 1.1009 | 0.0830 | 0.80 | 0.8299 | 0.0914 | 0.92 |

**Table S2.** Storage offsets ($S_o$) for near zero flow conditions of 0.01% and 0.1% of the minimum non-winter monthly discharge ($Q_{min}$) observed during the period of study and maximum and minimum basin-average GRACE TWSA observed during the period of study.

| ID | $S_o$ (0.1%$Q_{min}$) | $S_o$ (0.01%$Q_{min}$) | $TWSA_{max}$ | $TWSA_{min}$ |
|----|----------------------|------------------------|--------------|--------------|
| 1 | 84 | 106 | 12.5 | -18.0 |
| 2 | 79 | 100 | 14.0 | -17.9 |
| 3 | 114 | 145 | 15.2 | -20.5 |
| 4 | 93 | 119 | 10.7 | -14.0 |
| 5 | 84 | 107 | 10.9 | -13.8 |
| 6 | 84 | 108 | 12.6 | -14.4 |
| 7 | 145 | 190 | 18.9 | -7.9 |
| 8 | 158 | 207 | 20.8 | -7.2 |
| 9 | 168 | 220 | 21.3 | -7.6 |
| 10 | 146 | 191 | 20.1 | -8.1 |
| 11 | 111 | 144 | 18.4 | -7.7 |
| 12 | 96 | 123 | 14.6 | -9.7 |